# Blood-based epigenetic instability linked to human aging and disease

Salman Basrai[1,2], Ido Nofech-Mozes [1,2], Rajesh Detroja[3], Fernando L. Scolari[4,5], Mehran Bakhtiari[3], Andrea Arruda[3], Tracy Murphy[6], Scott V. Bratman[3,7], Steven M. Chan [3,8], Mark D. Minden[3,6,8], Jae Sook Ahn[9], Dennis D. H. Kim[3,6], Robert Kridel [3,8], Filio Billia [4,10] & Sagi Abelson [1,2] ✉

The abundance, dynamics, and context-dependent heterogeneity of DNA methylation, where a pattern considered abnormal in one cell type may be normal in another, complicate the identification of early methylation changes that drive or signal disease development. This complexity can obscure early markers of increased disease risk, making it challenging to detect and intervene in disease processes at their inception. Here, we report 31,744 CpG loci exhibiting highly consistent methylation profiles in the blood of young, healthy individuals. We assess alterations at these epigenetically stable loci in 8,886 individuals across 29 diverse cohorts, including those with hematological cancers (n = 3159), cardiovascular complications (n = 2788), and healthy controls (n = 2939). Our findings reveal methylation pattern disruption in myeloid and lymphoid malignancies, correlating with clonal burden dynamics and mutation frequency throughout leukemia treatment. In non-cancer cohorts, we observe that methylation levels at epigenetically stable loci become increasingly variable with age, a shift linked to higher cardiovascular disease risk and lower survival rates. This study highlights DNA methylation instability as a blood-based biomarker for both hematological cancer and cardiovascular disease and uncovers a mechanistic link between methylation dynamics and the expansion of maladaptive hematopoietic clones.

Clonal hematopoiesis (CH) of indeterminate potential is an age-related condition marked by the dominance of one or more genetically distinct clones of blood cells[1–3]. This phenomenon, often initiated by leukemia-associated gene mutations in hematopoietic stem and progenitor cells[4,5], is associated with increased risks of blood cancer development[6–9], adverse cardiovascular events[10,11], and overall mortality risk[6,12]. While the genetic basis of clonal hematopoiesis is well studied, the non-mutational mechanisms driving the emergence and

expansion of blood cell clones remain unclear. Recent studies have shown that 'driverless' clonal hematopoiesis, clonal expansion without known genetic drivers, is common and carries health risks comparable to cases with mutations in driver genes[6,13–15], underscoring the need to uncover additional mechanisms contributing to the reduced hematopoietic diversity observed in the elderly.

Somatic cell evolution is driven by genetic alterations that enable cells to circumvent normal regulatory mechanisms governing

---

[1]Ontario Institute for Cancer Research, Toronto, ON, Canada. [2]Department of Molecular Genetics, University of Toronto, Toronto, ON, Canada. [3]Princess Margaret Cancer Centre, University Health Network, Toronto, ON, Canada. [4]Peter Munk Cardiac Centre, University Health Network, Toronto, ON, Canada. [5]Hospital de Clínicas de Porto Alegre, Porto Alegre, Rio Grande do Sul, Brazil. [6]Division of Medical Oncology and Hematology, Princess Margaret Cancer Centre, Toronto, ON, Canada. [7]Department of Radiation Oncology, University of Toronto, Toronto, ON, Canada. [8]Department of Medical Biophysics, University of Toronto, Toronto, ON, Canada. [9]Chonnam National University Hwasun Hospital, Chonnam National University, Hwasun, Korea. [10]Department of Physiology, University of Toronto, Toronto, ON, Canada. ✉e-mail: SAbelson@oicr.on.ca

proliferation and tissue organizations[16,17]. For these alterations to enhance clonal fitness, they must be both persistent and heritable, thereby increasing the population of affected cells and facilitating the accumulation of additional drivers of clonal expansion. In extreme cases, the accumulation of genetic alterations can result in more extensive genomic instability and cancer[16,18]. Consistent with these established principles, genetic mutations follow a paradigm in which they persist throughout the lifespan of a cell and its lineage[19,20]. Conversely, epigenetic modifications present a paradox: while they can be heritable across cell divisions, they are also influenced by environmental factors and intrinsic cellular cues, making them more reversible and malleable than genetic mutations[21]. These features complicate efforts to define a clear baseline of epigenetic normalcy. In the absence of a stable reference framework, it becomes challenging to distinguish early or subtle pathological epigenetic changes from normal variation, limiting our understanding of how epigenomic alterations contribute to disease onset and progression.

Individuals can show considerable differences between their chronological years and their physiological state, reflecting variability in the rate of functional decline and susceptibility to disease. Among the most widely used biomarkers of this process are epigenetic clocks, which leverage DNA methylation (DNAm) patterns at CpG loci to estimate biological age. These clocks can capture aging-related changes, including associations with morbidity and overall mortality risk[22]. Clonal hematopoiesis has also been linked to epigenetic age acceleration[23], although correlations are generally weak, consistent with the notion that stochastic CpG epimutations are largely excluded from conventional clocks. Traditionally, epigenetic clocks are built using penalized multivariate regression, where a function of age is expressed as a weighted linear combination of CpG methylation values. An alternative approach emphasizes DNAm outliers rather than linear associations with chronological age[24,25]. In this framework, CpGs exhibiting the greatest deviations from standard methylation states are considered informative, offering complementary insight into biological aging processes. This approach has been especially valuable for cancer-risk prediction, with DNAm outliers in preneoplastic tissues marking subclonal expansions with increased malignant potential, as demonstrated in cervical and breast cancers[26,27]. Beyond oncology and certain neurodegenerative diseases[28], the full spectrum of disorders influenced by DNAm outliers and the degree to which these changes are truly stochastic remains unclear.

In this work, we investigate DNA methylation instability (DMI) in blood, examining a broad set of normally stable CpG loci. We find that DNAm variability at those loci increases with age and can inform leukemic burden and cardiovascular risk. We also explore mechanisms by which methylation of these sites may alter cellular function and contribute to disease, offering insights into the biological consequences of epigenetic instability.

## Results

### Low-variability CpG loci define a stable epigenetic landscape in human blood

To identify stable features of the human methylome, we analyzed whole blood DNAm profiles from a large cohort of young, healthy individuals[29] ($n = 1658$, age = 18; see Supplementary Data 1), in whom substantial clonal expansions are expected to be rare[1–3]. Genome-wide DNAm levels were assessed using Illumina 450 K Infinium arrays. Methylation levels were calculated using the ratio of intensities between the methylated and the combined locus intensity (β-value). CpG sites were restricted to autosomes, excluding sites near common single-nucleotide polymorphisms, sites with previously reported cross-reactivity[30], and sites predictive of biological age[31,32]. We defined Epigenetically Stable Loci (ESLs) as the 10% least variable CpG sites based on methylation level variance across the cohort. Given the heterogeneous cellular composition of peripheral blood, we validated

stable methylation levels across isolated blood cell populations and further refined our selection of ESLs by excluding CpG loci exhibiting outlier methylation levels (i.e., β-values) in any of the purified cell types. This approach identified 31,744 unmethylated (Supplementary Data 2) and 6143 methylated sites, with no sites exhibiting intermediate methylation levels (Fig. 1a, b; Supplementary Fig. 1). Notably, although these ESLs were identified in blood, we were able to detect their characteristic methylation patterns across non-hematological tissues using different profiling technologies (Fig. 1c). While some variation exists, the overall patterns, largely extreme hyper- or hypomethylation across ESLs, are consistently preserved across diverse tissue types (Supplementary Fig. 2a) and developmental stages (Supplementary Fig. 2b–d), suggesting the presence of stringent regulatory control established early in embryonic development and largely maintained throughout human life.

### Epigenetic perturbations at ESLs occur frequently across blood malignancies

We quantified DNAm at ESLs across a broad spectrum of myeloid and lymphoid malignancies to evaluate their disruption during carcinogenesis. To enable inter-cohort comparisons, we established a destabilization threshold based on ESL β-values from three independent control cohorts (total $n = 878$), representing baseline methylation levels at these sites (Fig. 2a; β-value = 0.05932). These cohorts encompassed diverse ethnic backgrounds, age ranges, and tissue sources, ensuring that the threshold reflects a broadly generalizable background level. Apart from chronic-phase CML, all investigated malignancies exhibited elevated methylation at normally unmethylated ESLs (Fig. 2a). This pattern was observed across both pediatric and adult cancers, independent of biopsy tissue origin or patient sex. Lymphoid cancers exhibited a greater degree of epigenetic destabilization than myeloid cancers, reflected in both the number of ESLs with β-values above the threshold established from healthy controls (termed perturbed ESLs) and in methylation levels at those typically hypomethylated loci (Fig. 2a, b). Reduced methylation of normally methylated ESLs was also observed, though this pattern was less consistent across cancers and more variable among controls (Supplementary Fig. 3). Accordingly, we focused on the unmethylated ESLs for all subsequent analyses; any mention of ESLs hereafter refers exclusively to this subset.

### Chromatin status in precursor cells associates with ESL perturbation susceptibility

Given the mechanistic link between DNAm and chromatin architecture[21,33,34], and the differing levels of ESL destabilization in cancers of distinct lineages, we examined chromatin accessibility at lineage-enriched, perturbed ESLs in corresponding cell types. To identify lineage-enriched perturbed ESLs, we compared AML profiles ($n = 997$) to T-ALL ($n = 353$) and BCP-ALL ($n = 663$). This analysis identified 1075 ESLs, predominantly perturbed in T-ALL and BCP-ALL, compared to 80 myeloid-enriched ESLs (Supplementary Data 2). This disparity aligns with the lower degree of ESL hypermethylation observed in AML relative to lymphoid cancers (Fig. 2). Single-cell ATAC-seq profiles of healthy hematopoietic cells were used to explore the relationship between instability in lineage-enriched perturbed ESLs and chromatin accessibility at these sites. Chromatin regions corresponding to lymphoid-enriched instability exhibited low accessibility in common lymphoid progenitors, T-cell, and B-cell lineages, and relatively higher accessibility in hematopoietic stem cells (HSCs), common myeloid progenitors, and granulocyte-monocyte progenitors. Conversely, myeloid-enriched perturbed ESLs exhibited significantly reduced chromatin accessibility in HSCs and myeloid progenitors, but relatively increased accessibility in lymphoid-lineage-associated cells (Fig. 3). These contrasting patterns link lineage-associated ESLs, their frequent perturbation in leukemia, and

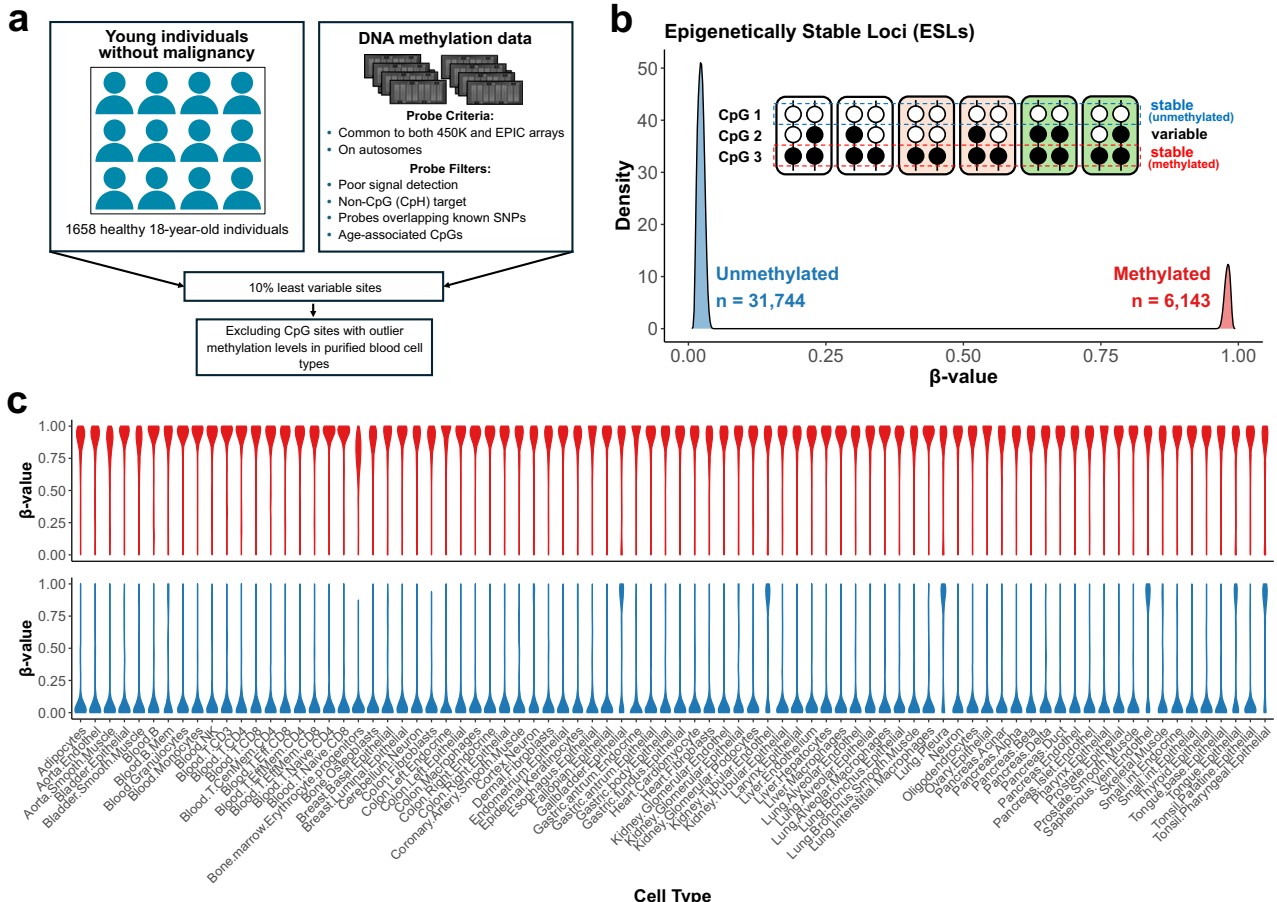

**Fig. 1 | Identification of epigenetically stable loci (ESLs) across the human genome. a** Schematic overview of the approach used to identify ESLs from blood DNA methylation data. **b** Density plot of average β-values for all identified ESLs. Blue shading indicates ESLs characterized by stable hypomethylation, while red shading represents ESLs characterized by stable hypermethylation. Inset: Illustration of 6 diploid cells from 3 individuals. CpG 1 is stably unmethylated, CpG 2 is variable across individuals, and CpG 3 is stably methylated. **c** Violin plots showing the distribution of β-values for ESLs identified in blood across 82 human cell types using whole-genome bisulfite sequencing data. Top panel: ESLs stably hyper-methylated in blood; bottom panel: ESLs stably hypomethylated in blood. Source data are provided as a Source Data file.

differential chromatin accessibility in corresponding precursor cells, an observation confirmed in a second dataset (Supplementary Fig. 4a–d). We further confirmed these associations using an orthogonal data modality (Supplementary Fig. 4e, f).

## Long-lasting methylation changes at ESLs indicate clonal epigenetic memory in leukemia

We curated and analyzed a pan-hematological cancer cohort of 3019 samples (Supplementary Data 1, 3) to assess the recurrence of ESL destabilization events. ESLs that were destabilized in a higher proportion of patients exhibited greater average methylation levels among those affected samples, suggesting that recurrent destabilization is associated with a larger fraction of methylated cells within each patient. (Fig. 4a, Supplementary Fig. 5). To evaluate somatic methylation inheritance at ESLs versus in-situ methylation occurring independently across cells, we focused on ESLs perturbed in fewer than 5% of patients, as these are more likely to represent patient-specific, clone-related epigenetic alterations. Analyzing paired diagnosis and relapse leukemia samples, we identified the 15 low-recurrence ESLs with the highest methylation levels in relapse samples for each patient. Unsupervised clustering then paired diagnosis samples with their corresponding relapse specimens. In AML ($n = 15$ diagnosis-relapse pairs), this approach resulted in 9 correct matches. For chronic lymphocytic leukemia (CLL), 31 out of 40 pairs were correctly paired. In BCP-ALL ($n = 24$ pairs), all diagnosis samples were correctly assigned to their

corresponding relapse sample (Fig. 4b–d). These results provide strong statistical evidence ($P = 0.002$ for AML; $P < 0.001$ for CLL and BCP-ALL by permutation test; Supplementary Note 1) for the clonal preservation of ESL methylation, suggesting long-term epigenetic memory in leukemia cells throughout disease progression. In patients with available remission samples, the observed reduction in DNAm supports a somatic, leukemia-specific signal (Fig. 4e). Importantly, the patient-specific backtracking of methylation at these ESLs from relapse to diagnosis indicates that the methylation observed at ESLs in therapy-resistant epigenetic clones (epi-clones) was established in the majority of patients prior to treatment initiation. Similarly, significantly improved pairing compared to random expectation was observed when analyzing the 15 most destabilized low-recurrence ESLs from diagnostic samples (Supplementary Fig. 6). Overall, these results support somatic and persistent methylation at ESLs, characteristics that are relevant for driving potential functional impact.

## DNA methylation instability is largely independent of genomic background

To quantify ESL perturbation on a per-sample basis, we introduced a metric called DNA methylation instability (DMI), defined as the standard deviation of β-values for ESLs within each sample. Since most ESLs remain unperturbed in individual samples, we focused on recurrently perturbed ESLs ($n = 9164$), defined as those perturbed in over 5% of samples in the pan-hematological cancer cohort

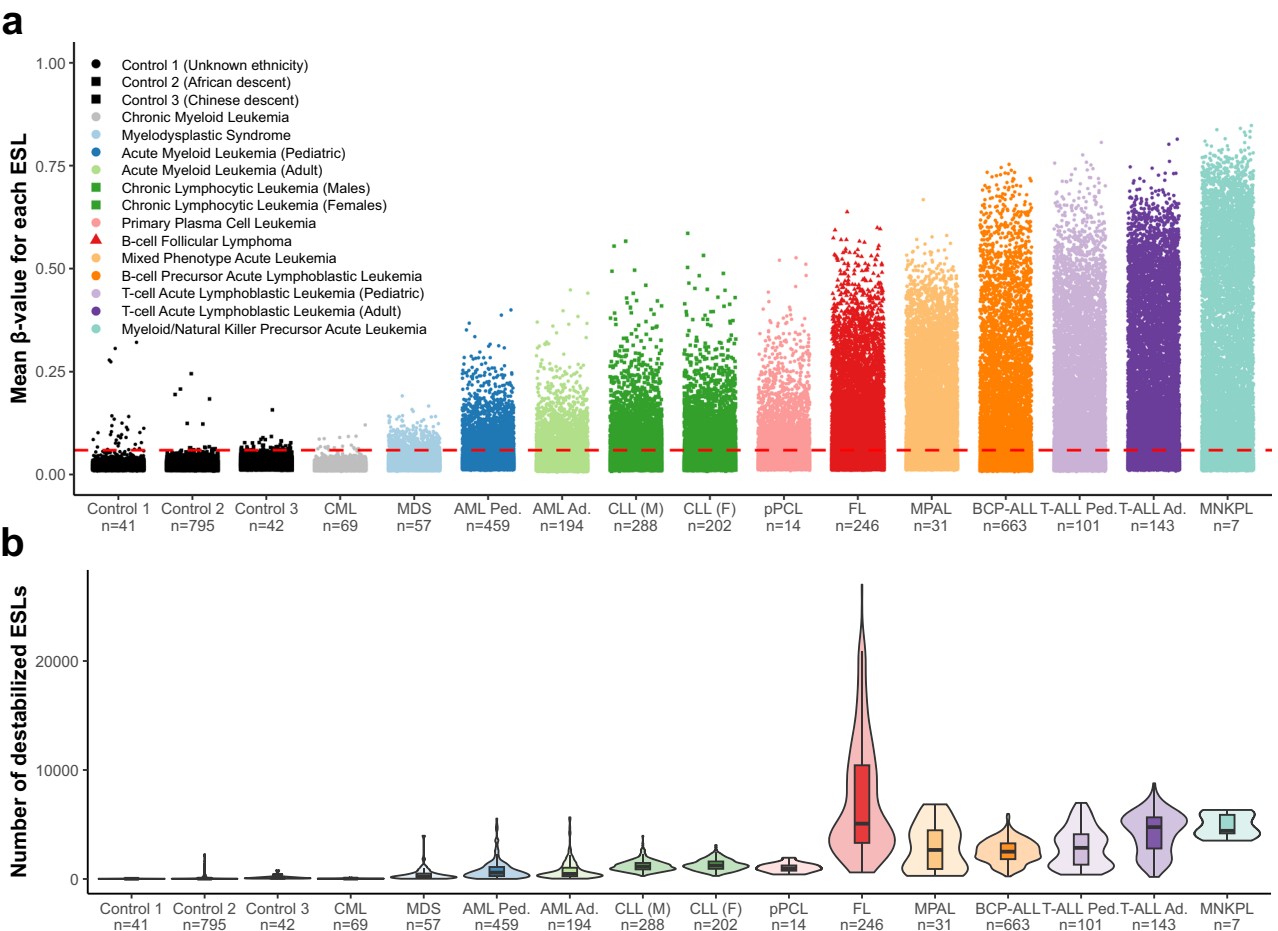

**Fig. 2 | Destabilization of unmethylated ESLs in hematological malignancies.**
**a** Average β-values of 31,744 unmethylated ESLs across various cancer and control cohorts. Each point represents the average β-value for a single ESL in a given cohort. The red dashed line indicates the destabilization threshold derived from the control samples. Marker shapes denote tissue source: circles = bone marrow, squares = peripheral blood, triangles = lymphoid tissue. **b** Boxplots and violin plots showing the number of destabilized ESLs per cohort. An ESL was considered destabilized if its β-value exceeded the 99.9th percentile of values observed in the control cohorts (destabilization threshold: β-value = 0.05932). For boxplots: black line represents the median, bounds of the box represent the first and third quartiles, whiskers extend to (median ± 1.5 * interquartile range), and points represent outliers. CML chronic myeloid leukemia, MDS myelodysplastic syndrome, AML acute myeloid leukemia, CLL chronic lymphocytic leukemia, pPCL primary plasma cell leukemia, FL follicular lymphoma, MPAL mixed phenotype acute leukemia, BCP-ALL B-cell precursor acute lymphoblastic leukemia, T-ALL T-cell acute lymphoblastic leukemia, MNKPL myeloid/natural killer cell precursor acute leukemia, Ped pediatric, Ad adult, M male, F female. Source data are provided as a Source Data file.

(Supplementary Data 3). Longitudinal measurements of bone marrow samples from AML patients revealed that DMI levels correspond to expected patterns of leukemic burden across clinical stages of the disease (Fig. 4f), consistent with our observations of somatic DNAm modifications based on individual ESLs in BCP-ALL (Fig. 4e). Notably, in Patient 2, who was clinically defined as being in remission at the third sampling, we observed an increase in DMI, potentially indicating the impending relapse, which was diagnosed 24 days later. DNAm profiling of a subset of leukemia patients with distinct genetic mutations, previously characterized by next-generation sequencing[35], revealed that DMI levels closely reflected the allelic burden of these mutations (Fig. 4g, Supplementary Fig. 7). Notably, in Patient 8, where clonal tracking was hindered by the absence of identifiable mutations, the ability to monitor DMI highlights its potential as a complementary approach to traditional mutation-based methods for assessing complete remission and residual disease post-treatment.

### Progressive epigenetic instability with age
Given that DMI measurements share similarities with genetic clonal markers in the hematopoietic system, particularly in their association with leukemic cell burden and the persistence of the alteration, and

considering that hematopoietic clonality increases in frequency with age[36–38], we hypothesized that DMI may also reflect age-related changes in individuals without hematological malignancies. To test this relationship, we examined several cohorts of healthy blood donors and found a significant correlation between DMI and age (Fig. 5). These results demonstrate that global deviation from the normally hypomethylated state of ESLs increases progressively, suggesting that DMI may be a general feature of aging hematopoietic cells rather than a phenomenon restricted to malignant transformation. Notably, DMI levels in these healthy cohorts were consistently lower than those measured in individuals with hematological malignancies (Fig. 4f, g, Supplementary Fig. 7, 8), suggesting an accelerated rate of methylation instability following malignant transformation.

### DNA methylation instability is associated with increased cardiovascular risk and mortality
Clonal hematopoiesis has consistently been linked to increased cardiovascular risk and related adverse outcomes[10,11,39–41]. Given the similar associations between genetically defined clonality and DMI in cancer and aging, we hypothesized that elevated DMI could serve as an indicator of cardiovascular disease. To investigate this relationship, we

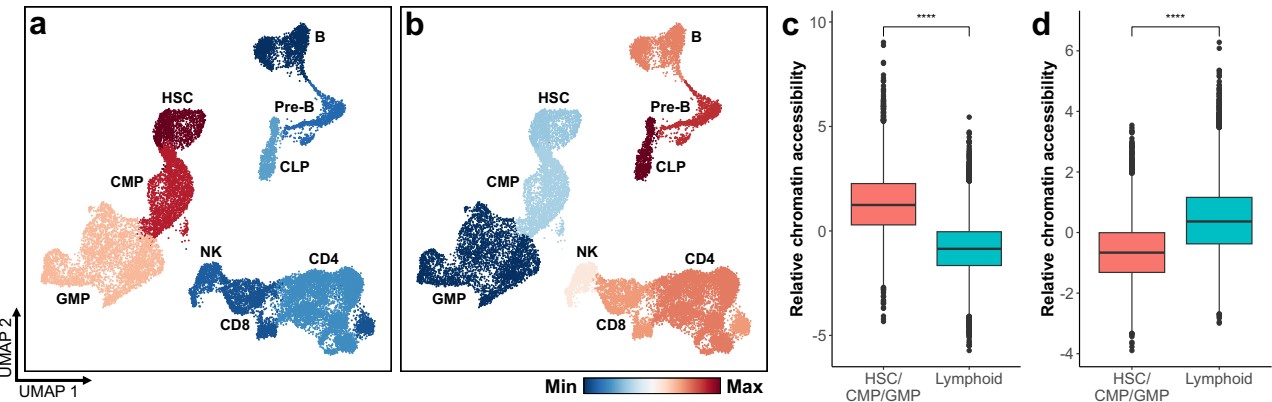

**Fig. 3 | Chromatin accessibility patterns in normal hematopoietic cells at lineage-enriched perturbed ESLs. a**, **b** UMAP projections of single-cell ATAC-seq profiles from normal blood cells (*n* = 22,737). Cells are colored by their relative chromatin accessibility at genomic regions overlapping **a** lymphoid-enriched or **b** myeloid-enriched ESLs. Accessibility was quantified using deviation scores that reflect enrichment relative to a background model. Red indicates higher relative accessibility (less compacted chromatin), and blue indicates lower relative accessibility (more compacted chromatin). **c**, **d** Relative chromatin accessibility at ESL-associated regions in hematopoietic stem and progenitor cells (HSCs, CMPs, GMPs; *n* = 8321) compared with lymphoid cell types (CLP, Pre-B, B, CD4⁺ T, CD8⁺ T, NK;

*n* = 14,416), for **c** lymphoid-enriched ESLs and **d** myeloid-enriched ESLs. For box-plots: black line represents the median, bounds of the box represent the first and third quartiles, whiskers extend to (median ± 1.5 * interquartile range), and points represent outliers. ****$P < 2.2 \times 10^{-16}$ by two-sided Wilcoxon rank-sum test. ATAC-seq assay for transposase-accessible chromatin using sequencing; UMAP uniform manifold approximation and projection, HSC hematopoietic stem cell, CMP common myeloid progenitor, GMP granulocyte-monocyte progenitor, CLP common lymphoid progenitor, NK natural killer. Source data are provided as a Source Data file.

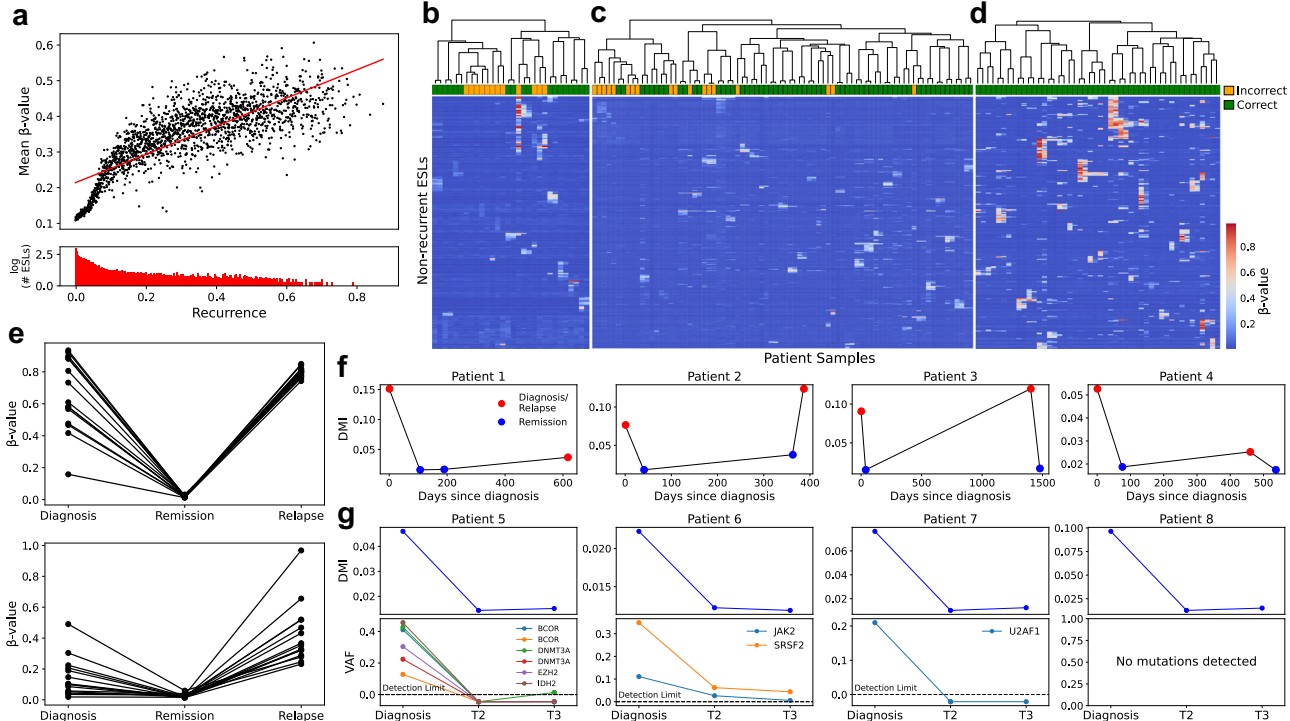

**Fig. 4 | Aberrant DNA methylation at ESLs links leukemic epigenetic clones at diagnosis and relapse. a** Recurrence of ESL perturbation is shown on the x-axis as the proportion of patients (*n* = 3019) in which a given locus exceeds the methylation destabilization threshold. The y-axis represents the mean β-value calculated only across patients in whom that locus is perturbed. Thus, the plot reflects the relationship between the frequency of ESL destabilization across individuals and the average methylation magnitude among those affected samples. Pearson's correlation coefficient (r) = 0.79; *P* < 0.001 by two-sided *t*-test. The bottom panel shows the number of ESLs (log₁₀ scale) across different recurrence levels. Hierarchical clustering of paired diagnosis and relapse samples: **b** 15 AML patients, **c** 40 CLL patients, and **d** 24 BCP-ALL patients. The 15 low-recurrence sites (i.e., perturbed in <5% of the pan-hematological cancer cohort) with the highest methylation levels

were selected from each patient's relapse sample for clustering. Correctly paired diagnosis-relapse samples are indicated in green. *P* = 0.002 by two-sided permutation test for AML; *P* < 0.001 for CLL and BCP-ALL. **e** β-values of the top 15 low-recurrence ESLs in two BCP-ALL patients sampled at diagnosis, remission, and relapse. Each line connects the β-values of a single ESL across disease stages. **f** DMI levels in bone marrow samples from four AML patients, sampled at four distinct time points during disease progression. **g** Analysis of peripheral blood samples from four additional AML patients at diagnosis and two subsequent time points during remission. The top panels display DMI levels, and the bottom panels show VAFs of the patients' leukemic mutations. VAF variant allele frequency. Source data are provided as a Source Data file.

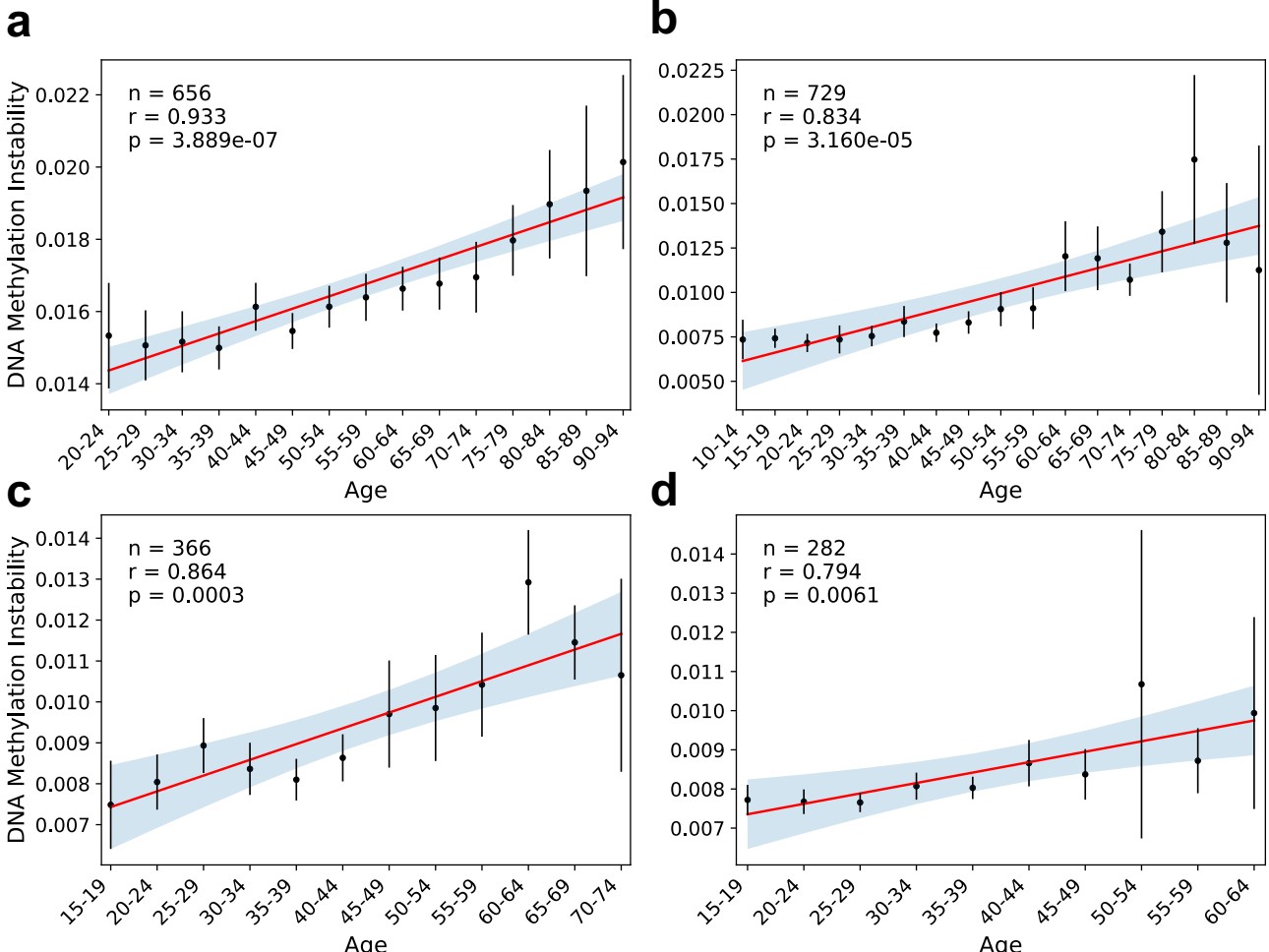

**Fig. 5 | Increasing DNA methylation instability correlates with advanced age.** DMI levels were measured across four independent cohorts: **a** GSE40279, **b** GSE87571, **c** GSE115278, **d** GSE197676. Participants were grouped into 5-year age intervals. Black dots represent the mean DMI for each age group, with error bars indicating the 95% confidence interval. Linear regression was performed on group means; the regression line is shown in red, with the 95% confidence interval shaded in blue. The number of samples, Pearson's correlation coefficients (*r*), and two-sided *P*-values are indicated. Source data are provided as a Source Data file.

analyzed DNAm data generated as part of the Framingham Heart Study[42], with follow-up spanning up to 14 years. Kaplan–Meier analysis showed that individuals with elevated DMI had a significantly higher risk of all-cause mortality, as well as cardiovascular disease (CVD), coronary heart disease (CHD), and congestive heart failure (CHF) compared to those with low DMI (Fig. 6a–d). Hazard ratios for elevated CVD and CHD risk remained significant after adjusting for age and sex (Fig. 6e). Previous analyses of health outcomes revealed that higher fractions of naïve CD4+ T cells, naïve B cells, and NK cells were associated with a reduced risk of all-cause mortality, independent of major epidemiological risk factors and baseline comorbidities[43]. Cox regression analyses adjusting for the relative abundance of immune cell types[44] confirmed that the associations between high DMI and CVD and CHD risk remained significant (Supplementary Data 4). To assess the impact of extreme epigenetic profiles, we repeated the analyses considering only individuals in the top and bottom quartiles of DMI values. This comparison of extreme quartiles accentuated the differences in clinical outcomes between individuals with high versus low DMI (Supplementary Fig. 9).

We previously performed next-generation sequencing on cardiogenic shock patients and identified significant survival differences between those with clonal hematopoiesis and those without detectable clonal hematopoiesis[39]. To investigate the prognostic value of DMI in these high-risk patients, we profiled a subset of the original

cohort, including 29 patients with clonal hematopoiesis driven by common *DNMT3A* or *TET2* mutations, and 35 without clonal hematopoiesis. As a result, we intentionally obtained an underpowered cohort with respect to the significant association between clonal hematopoiesis and mortality rates seen in our original study[39] (Supplementary Fig. 10). Despite this, individuals with elevated DMI had significantly lower survival compared to those with low DMI (Fig. 6f). This association remained significant after adjustment for age and clonal hematopoiesis status (Cox proportional hazards models: HR = 3.76; *P*-value = 0.006), and after accounting for immune cell fractions (Supplementary Data 4). Stratification of patients based on DMI levels and clonal hematopoiesis status demonstrated that DMI is an independent risk factor (Fig. 6g). This finding further supports our earlier observation that elevated DMI is largely independent of the underlying genomic context (Fig. 4g, Supplementary Fig. 7).

### ESLs overlap transcription factor motifs and are linked to gene silencing

The widespread associations between DMI and aging, cancer, and cardiovascular disease prompted us to investigate underlying mechanisms. Given DNA methylation's role in gene regulation, we hypothesized that accumulating epigenetic instability may disrupt gene expression. Consistent with this, ESLs were significantly enriched within CpG islands relative to non-ESLs (OR = 15.1, *P* < 0.001) and were

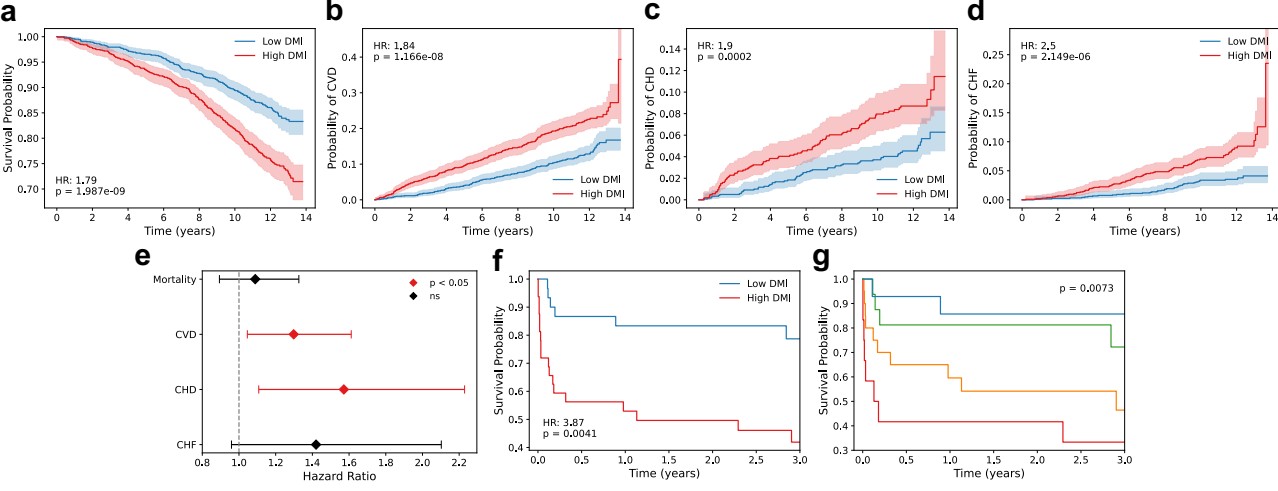

**Fig. 6 | Association of DNA methylation instability with cardiovascular risk and mortality.** Univariate Kaplan–Meier analyses of 2281 participants from the Framingham Heart Study (Exam 8) for **a** mortality, **b** cardiovascular disease, **c** coronary heart disease, and **d** congestive heart failure. Participants were classified into low or high DMI groups based on the cohort median split. Shaded areas represent the 95% confidence interval. Hazard ratios (HR) and *P*-values are indicated. **e** Forest plot showing the effect of high DMI on each endpoint using multivariate Cox proportional hazards regression on the same 2281 samples, adjusted for age and biological sex. Error bars represent the 95% confidence interval. Mortality: HR = 1.09, *P* = 0.400; CVD: HR = 1.30, *P* = 0.018; CHD: HR = 1.57, *P* = 0.011; CHF: HR = 1.42, *P* = 0.080. **f** Survival analysis of cardiogenic shock patients (*n* = 64), stratified into low DMI or high DMI groups based on the cohort median split. **g** Survival curves of the same patients, further stratified into the following categories: i) high DMI with CH mutations, ii) high DMI without CH mutations, iii) low DMI with CH mutations, and iv) low DMI without CH mutations. *P*-values were calculated using the multivariate log-rank test. CVD cardiovascular disease, CHD coronary heart disease, CHF congestive heart failure, CH clonal hematopoiesis. Source data are provided as a Source Data file.

preferentially located in gene promoters, with the majority (92.6%) occurring within 1500 bp of a transcription start site (TSS) (Fig. 7a, b).

To identify genes most likely influenced by ESL perturbation, we focused on three criteria (Fig. 7c): the presence of recurrently methylated promoter-associated ESLs, as infrequent perturbation and lower methylation (Fig. 4a) are less likely to drive meaningful regulatory changes; a negative correlation between ESL methylation levels and gene expression; and a negative correlation between gene expression and age. Age-related genes were identified using data from the Genotype-Tissue Expression project[45] (Supplementary Fig. 11), while the Cancer Genome Atlas data[46], which provides paired methylation and expression profiles across large, heterogeneous cancer cohorts, was used to identify genes for which promoter ESL methylation was associated with lower gene expression ("Methods" section). Genes meeting all three criteria were prioritized for further investigation (*n* = 236; Supplementary Data 5).

*PRDM5* is a well-characterized tumor suppressor frequently silenced by promoter hypermethylation in multiple solid cancers, the loss of which is associated with increased tumor cell proliferation and poor prognosis[47–49]. Among all genes, the *PRDM5* promoter harbors ESLs with the highest likelihood of perturbation in the pan-hematological cancer cohort (Supplementary Data 5), reinforcing independent evidence of widespread CpG island hypermethylation at this locus observed in AML (Supplementary Fig. 12a). A similar trend towards widespread promoter hypermethylation in AML was observed across the 236 prioritized genes (Fig. 7d, Supplementary Data 5, Supplementary Fig. 12b, c).

Functional enrichment analysis revealed significant enrichment of transcription factor (TF) binding motifs within the 236 genes, including those for the *SP* and *AP-2* families, linking ESL perturbation to key regulators of the cell cycle, differentiation, proliferation, and apoptosis[50,51] (Supplementary Data 6). An independent analysis confirmed that the CpG loci corresponding to ESLs are integral components of many of the enriched motifs (Fig. 7e). A notable example is *RGS2*, whose promoter region contains recurrently perturbed ESLs that overlap with several TF motifs (Fig. 7f). Reduced expression of

*RGS2* in peripheral blood mononuclear cells has been associated with hypertension[52]. Given that these cells are known to exhibit functional abnormalities in hypertension[53] and contribute to the pathogenesis of atherosclerosis[10], *RGS2* dysregulation in blood through ESL perturbation may contribute to the observed association with cardiovascular disease, although a causal role remains to be established.

To shed more light on the potential role of ESL methylation in modulating TF binding, we cross-referenced promoter-proximal ESLs (Fig. 7e) with TF motifs whose methylation sensitivity has been characterized using methylation-sensitive SELEX[54]. This approach allowed us to identify TFs for which the presence of 5-methylcytosine (5mC) within the binding motif can either enhance or inhibit binding (Supplementary Data 7). KLF12 emerged as a compelling example: its binding is enhanced by 5mC, and it functions as a transcriptional repressor[55,56]. Perturbation of ESLs within KLF12 target promoters (Fig. 7f) could therefore influence gene expression by altering TF occupancy. We emphasize that the impact of ESL methylation on gene regulation is inherently complex. The functional consequence depends not only on whether 5mC enhances or inhibits TF binding, but also on the TF's regulatory role as activator, repressor, or context-dependent modulator, and on combinatorial effects when multiple TFs and ESLs are involved.

Together, these examples illustrate a broader mechanism by which ESL perturbations may modulate the binding of common TFs at promoter regions, thereby altering the expression of age-related and disease-associated regulatory genes critical for maintaining hematopoietic homeostasis.

## Discussion

In this study, we identify epigenetic instability as a risk factor associated with several hallmarks of clonal hematopoiesis. Our analysis identified a set of CpG loci, termed epigenetically stable loci (ESLs), that exhibit a consistently hypomethylated state across healthy young adults and diverse blood cell types. This stability extends to other normal tissues, suggesting that ESL methylation patterns are established during early development and largely maintained throughout

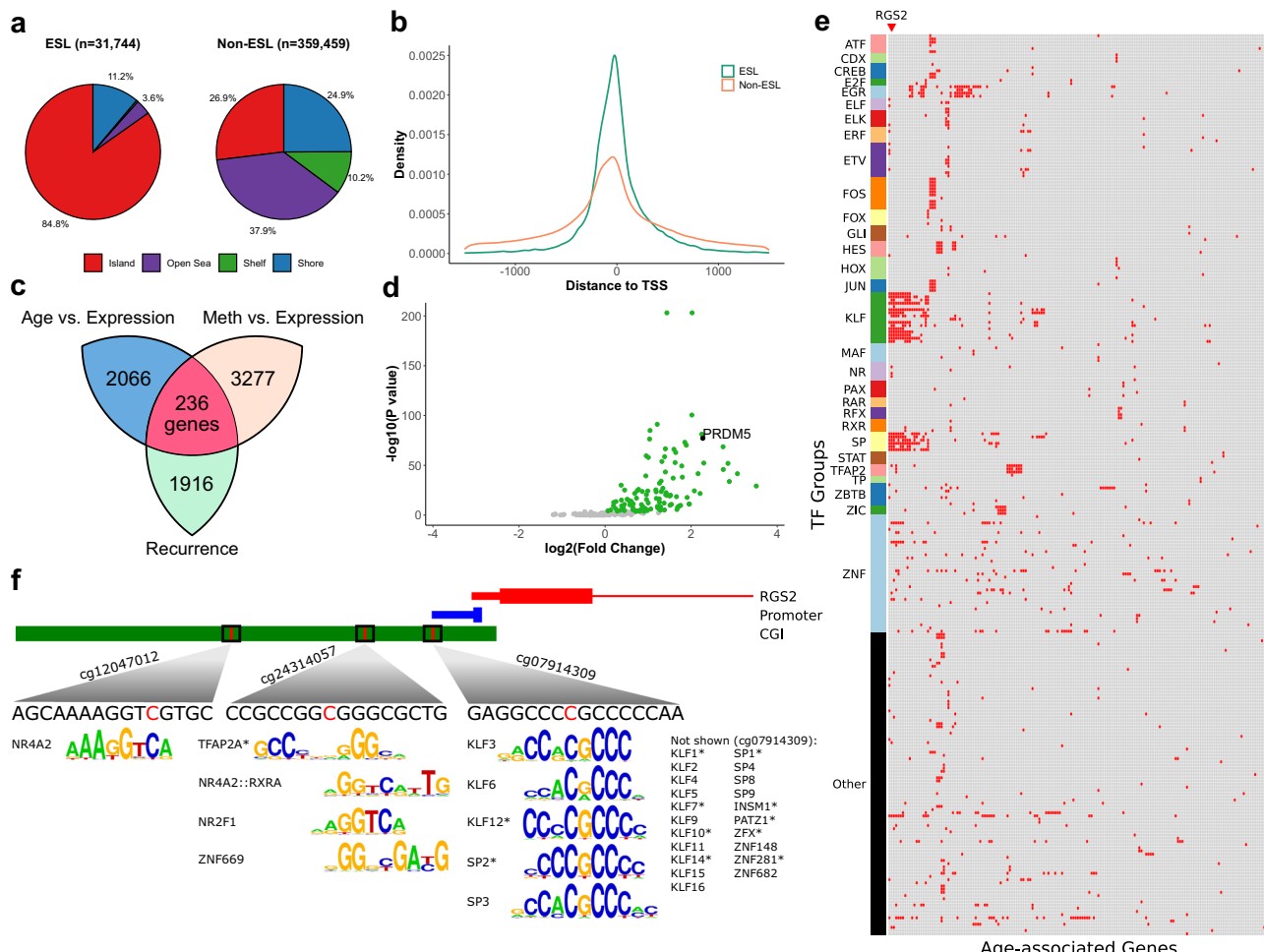

**Fig. 7 | DNA methylation instability at promoter regions and its impact on gene expression. a** Genomic distribution of ESLs (left) and non-ESLs targeted by the Infinium array (right), indicating enrichment of ESLs at CGI. $P < 0.001$ by two-sided Fisher's exact test. Each site is classified by its proximity to CGIs: within a CGI, in a shore (0–2 kb from an island), in a shelf (2–4 kb), or in the open sea (≥4 kb). **b** Density plot showing the distance to the nearest TSS for ESLs (green) and non-ESLs (orange). **c** Diagram depicting the total number of genes with recurrent ESLs in their promoter regions (i.e., recurrence greater than 5%; $n = 1916$), genes with a significant negative correlation between methylation and expression ($n = 3227$), genes with a significant negative correlation between age and expression ($n = 2066$), and the final intersected list of candidate genes ($n = 236$). **d** Volcano plot showing differential methylation analysis of promoter-associated CGIs for the 236 age-dependent genes. The y-axis represents the negative log-transformed $P$-values

(two-sided, Bonferroni-corrected), and the x-axis shows the $\log_2$ fold change. Significant increases in overall promoter methylation in AML compared to $CD34^+$ controls ($P < 0.05$ by two-sided Wilcoxon rank-sum test after Bonferroni correction) are highlighted in green. **e** Heatmap showing the overlap between age-dependent genes and transcription factor motifs within their promoter-associated CGI. Red cells highlight regions where the ESLs intersect with key transcription factor binding motifs. **f** Visualization of genomic features upstream of *RGS2*. The CGI, promoter, and first exon are shown. For the promoter element, the "thin" section represents the 49 bp region upstream of the TSS, while the "thick" section represents the TSS plus 10 bp downstream. Red lines on the CGI indicate ESLs. Transcription factor binding motifs containing the ESLs are shown. Asterisks denote reverse complement sequences. CGI CpG island, TSS transcription start site, TPM transcripts per million. Source data are provided as a Source Data file.

differentiation, consistent with current understanding of methylation reprogramming during embryogenesis[57].

We found that disruptions in the hypomethylated state of ESLs, accompanied by increased methylation variability, termed DNA methylation instability (DMI), are linked to adverse health outcomes. Our findings underscore the critical role of maintaining ESL hypomethylation in preserving stable and healthy hematopoiesis. ESL destabilization was seen across a wide range of hematological malignancies, with elevated DMI at leukemia diagnosis and significant reduction post-induction therapy, mirroring leukemic mutations and supporting somatic acquisition of this signal. Notably, elevated DMI was also observed in the absence of known driver mutations, suggesting its potential as a complement to genetic testing for enhanced diagnostic and prognostic assessment. DMI patterns persisted from diagnosis to relapse, indicating an epigenetic memory that endures selective pressures of cancer treatment. This persistence underscores

the heritability of ESL methylation during cell division and its potential to accumulate over time, priming healthy cells for malignant transformation.

DNA methylation can reflect both changes in clonal composition and functional epigenetic alterations occurring within individual clones. In the first scenario, clones harboring specific epigenetic variants may expand; in this case, stochastic ESL methylation marks would primarily serve as tracers of clonal dynamics and, on their own, are unlikely to directly alter cellular behavior[58,59]. Similar patterns are observed for other molecular markers, such as mtDNA mutations, which reveal a rich polyclonal architecture of hematopoietic stem cell contributions to hematopoiesis, with clonal expansions arising from multiple independent origin[60]. Alternatively, instability may occur within individual clones, progressively altering the epigenome and programmatically influencing gene regulation and cellular fitness, potentially leading to disease-associated patterns such as focal hyper-

or hypomethylation, disrupted TF networks, and altered clonal selection dynamics. Distinguishing these scenarios will help determine whether specific epimutations act as drivers of cellular behavior, representing actionable therapeutic targets, or as passengers reflecting clonal history, which can serve as robust biomarkers for diagnosis, prognosis, or monitoring treatment response.

We observed that elevated DMI levels correlate with age, reflecting the progressive nature of ESL methylation instability. Additionally, ESL perturbation was associated with differential chromatin accessibility, suggesting that ESL destabilization is not entirely stochastic, with chromatin context potentially predisposing specific loci to epigenetic disruption or influencing the stability of methylation once it occurs.

Similar to clonal hematopoiesis associations, elevated DMI was linked to increased cardiovascular risk and higher mortality in high-risk cardiac patients. We observed a hazard ratio for developing coronary heart disease similar to that reported for CH in other epidemiological cohorts[10,61]. Importantly, the association between DMI and survival among cardiogenic shock patients not only remained independent of CH mutations in *TET2* and *DNMT3A*, but also amplified their prognostic impact[39], suggesting that DMI contributes to cardiovascular risk through mechanisms that are both distinct from and additive to those of established CH drivers. Future studies in larger cohorts are needed to assess the independence of DMI from other CH mutations and its clinical utility across diverse pathological contexts.

Our findings link DMI to genes exhibiting age-associated downregulation that are involved in cell fitness, cancer, and cardiovascular disease. Many of these genes may be affected by ESL perturbations through their embedding in TF binding motifs. ESL perturbations, often observed in leukemia, coincide with *SP*, *KLF*, *FOS*, *JUN*, *EGR*, and *AP-2* motifs, suggesting that DMI may globally disrupt cellular processes involved in proliferation, apoptosis, and differentiation. As we age, this accumulated dysregulation could impair hematopoietic stem cell maintenance, lineage commitment, and increase disease susceptibility.

This study establishes a framework linking epigenetic stability to hematological normalcy and instability to reduced cellular diversity and pathogenesis. A limitation is the use of Infinium array data, which covers less than 2% of the human genome and is biased toward promoter regions. This underscores the need for more comprehensive profiling to explore intergenic, stable genomic regions that, when methylated, may act as disease drivers. Such profiling could uncover new regulatory regions with critical roles in disease progression. Future efforts to establish a comprehensive reference epigenome of normalcy will be essential for advancing our understanding of epigenetic drivers and mitigating the impact of both clonal and epi-clonal hematopoiesis on human health.

## Methods

### Ethics
Patient samples were collected in accordance with procedures approved by the Research Ethics Board of the University Health Network (REB #01-0573) and viably frozen in the Princess Margaret Leukemia Tissue Bank. Blood biospecimens from cardiogenic shock patients were collected under UHN REB approval (#18-6188) from the Peter Munk Cardiac Centre Cardiovascular Biobank. Written informed consent was obtained from all patients in accordance with the Declaration of Helsinki. University of Toronto REB approval was obtained for the use of datasets in secondary data analysis under Protocol #00041924.

### DNA methylation profiling
Venous blood and bone marrow biopsies were collected from 14 AML patients, resulting in a total of 46 longitudinal samples. For patients experiencing cardiogenic shock ($n = 64$), buffy coat samples were utilized for DNA methylation profiling. Genomic DNA was extracted using the QIAamp DNA Blood Mini Kit (Qiagen) according to the

manufacturer's instructions. The concentration and purity of the DNA were assessed spectrophotometrically on a Qubit Fluorometer, with criteria set at an A260:A280 ratio greater than 1.7 and an A260:A230 ratio greater than 1.7. DNA integrity was evaluated through 1% agarose gel electrophoresis, loading 50–100 ng of DNA per sample. Samples exhibiting a clear single band above the 10 kb DNA ladder were deemed suitable for further processing. High-quality genomic DNA (500 ng) was bisulfite-converted using the EZ DNA Methylation Kit (Zymo Research) following the manufacturer's protocol. DNA methylation profiling was conducted using the Infinium MethylationEPIC microarray (Illumina).

### Methylation array data preprocessing
Raw IDAT files were processed using the minfi R/Bioconductor package[62]. IDATs were loaded via read.metharray.exp into RGChannelSet objects. To ensure cross-platform consistency among different datasets, only probes shared between the 450 K and EPIC arrays were retained. Background correction and dye-bias normalization were performed using the preprocessNoob method, which applies single-sample normalization independently to each array[63]. β-values (0–1 methylation levels) were calculated using getBeta. We then applied additional probe filtering criteria. Probes with detection $P > 0.001$ in more than 5% of samples were removed using the detectionP function. We also excluded non-CpG (CpH) targeting probes, probes overlapping SNPs (identified with dropMethylationLoci and dropLociWithSnps, with no minimum minor allele frequency cutoff), probes with previously reported cross-reactivity[30], age-associated CpGs[31,32], and probes on sex chromosomes. A final set of 397,346 probes was considered for further analysis.

### Identification of epigenetically stable loci (ESLs)
Whole blood DNA from 1658 young individuals (age 18), sourced from Hannon et al.[29], was used to identify CpG sites with low variability in DNAm levels. The CpGs retained after data preprocessing ($n = 397,346$) were ranked by β-value variance, and the 10% least variable sites across the cohort were classified as epigenetically stable loci (ESLs). The average β-values at these stable sites showed a pronounced bimodal distribution, with one peak near zero (maximum β-value = 0.043) and another near one (minimum β-value = 0.962). Sites with low β-values were classified as unmethylated ESLs ($n = 31,744$, Supplementary Data 2), and those with high β-values as methylated ESLs ($n = 6143$). To account for potential residual cell type bias, β-values were assessed in purified blood cell types (B cells, CD4+ T cells, CD8+ T cells, granulocytes, and monocytes). ESLs showing outlier methylation levels in any of these cell types were excluded. In addition, we evaluated these ESLs in a separate dataset generated using whole-genome bisulfite sequencing from 82 purified human adult healthy tissue cell populations, sourced from Loyfer et al.[64]. In this dataset, β-values for each locus were calculated as the ratio of methylated reads to total reads.

### Establishing a threshold for epigenetically stable loci (ESL) destabilization
To determine whether an ESL was destabilized in a given patient, we established a threshold based on the β-values of ESLs observed across three control cohorts from the Gene Expression Omnibus (datasets: GSE124413, GSE132203, and GSE141682). For each control cohort, the mean β-value for each ESL was calculated individually, and all values were pooled. The destabilization threshold was set at the 99.9th percentile of the pooled values. This resulted in a threshold of β-value = 0.05932 for normally unmethylated ESLs. ESLs with methylation levels exceeding this threshold were classified as perturbed, also referred to as destabilized. Following the same approach, a β-value of 0.8321 was calculated for normally methylated ESLs (Supplementary Fig. 3).

## Frequency and recurrence of ESL destabilization in a pan-cancer hematological dataset

To assess the frequency of ESL destabilization, we compiled a pan-cancer dataset comprising 3019 samples from various hematological malignancies, including AML, T-ALL, BCP-ALL, CLL, FL, DLBCL, pPCL, MNKPL, and MPAL (Supplementary Data 1). CML and MDS were excluded from this analysis due to their characteristic low burden of blast cells and minimal ESL destabilization. A β-value matrix was generated, with CpGs as rows and patients as columns. This matrix was then binarized by assigning a value of 1 to β-values exceeding the predetermined ESL destabilization threshold (β-value = 0.05932), and a value of 0 to β-values below the threshold. For each ESL, the proportion of patients exhibiting destabilization was calculated by dividing the sum of the binarized values for that row by the total number of samples ($n = 3019$). Low-recurrence ESLs were defined as those perturbed in less than 5% of the samples, while recurrently perturbed ESLs were defined as those exhibiting perturbation in 5% or more of the samples (Supplementary Data 2). In the analysis examining the correlation between ESL perturbation recurrence and their average destabilized β-value (Fig. 4a, Supplementary Fig. 5), the average β-value for each ESL was calculated only from patients exhibiting destabilization at that site.

## Identification of lymphoid- and myeloid-enriched perturbed ESLs

A comparison was conducted between AML samples and T-ALL and BCP-ALL to identify ESLs whose destabilization shows a stronger association with myeloid or lymphoid leukemias. The matrix of binarized β-values was used to conduct a two-sided Fisher's test to assess the number of destabilized samples among myeloid cancer patients compared to lymphoid cancer patients. We only retained ESLs with a *P*-value < 0.05 following Bonferroni correction. To identify lymphoid-enriched perturbed ESLs, we first selected sites that had an odds ratio greater than 1 and exhibited destabilization in at least 50% of the combined T-ALL and BCP-ALL samples, thereby emphasizing ESLs that are frequent in these lymphoid leukemias. This set of ESLs was ranked by their odds ratios, and the top 50% of these sites were designated as lymphoid-enriched ESLs – those most strongly associated with lymphoid cancers. To identify myeloid-enriched perturbed ESLs, we adapted this approach to account for the relatively lower frequency of ESL perturbations observed in AML. Comparisons between AML and T-ALL, as well as AML and BCP-ALL, were conducted separately, and the intersection of both the resulting lists was designated as myeloid-enriched. Additionally, we retained only ESLs that exhibited destabilization in at least 10% of the AML samples.

## Chromatin accessibility and its association with lineage-enriched perturbed ESLs

A cell-by-peak matrix was obtained from Granja et al.[65], corresponding to peaks called from the aligned scATAC-seq fragments for each cell. In the original study, latent semantic indexing (LSI) followed by singular value decomposition (SVD) was used to perform dimensionality reduction, normalization, and selection of informative peaks. We leveraged this data to investigate the relationship between lineage-enriched perturbed ESLs and chromatin accessibility. Specifically, a subset of the LSI-SVD matrix was created, including the following relevant cell types: HSC, CMP, GMP, CLP, Pre-B cells, B cells, CD4 T cells, CD8 T cells, and NK cells. UMAP was performed on this matrix using the implementation in the uwot R package, with the same parameters used in the original publication. To assess chromatin accessibility at the lymphoid-enriched and myeloid-enriched ESLs, we first mapped ESLs to peaks in the scATAC-seq dataset by constructing, for each ESL, a 200 bp window with the CpG bases at the center. Any scATAC-seq peaks overlapping with these windows were retained. This process was performed separately for lymphoid-enriched ESLs ($n = 1075$) and myeloid-enriched ESLs ($n = 80$), yielding two peak sets corresponding to these loci. ChromVAR deviation scores[66] were generated for each peak set using the 'AddChromatinModule' function from the Signac R package[67]. Each cell type in the UMAP was colored according to its average chromVAR deviation scores for lymphoid-enriched ESLs and myeloid-enriched ESLs. The same approach was used for the analysis of a second single-cell ATAC-seq dataset from Izzo et al.[68]. In this dataset of hematopoietic cells derived from patients with JAK2[V617F]-mutated myelofibrosis, only cells that were wild type for this allele were analyzed ($n = 6088$). UMAP coordinates provided in the original publications were used for visualization.

## Power calculation for the selection of cardiogenic shock patients

Prior to DNA methylation profiling and the analysis assessing the impact of DMI status on the survival time of cardiogenic shock patients, we conducted a power analysis to determine the required sample size of a sub-cohort that would provide sufficient statistical power to detect a significant effect. This analysis was performed using the 'ssizeEpiCont' function from the powerSurvEpi R package. Over 10,000 iterations, we randomly assigned a fixed number of patients from the entire cardiogenic shock cohort from our original study[39] to high DMI status, while an equal number were assigned to low DMI status, mimicking the process of splitting patients based on median DMI values. The analysis was designed to achieve 85% power, assuming a hazard ratio of 1.5 and a type I error rate (alpha) of 0.05. This simulation indicated that a sub-cohort of 64 patients would be adequately powered to examine the relationship between DMI status and survival time in these high-risk patients.

## Survival and time-to-event analyses

DNA methylation instability (DMI) was defined as the standard deviation of ESL β-values within a sample. Participants in the Framingham Heart Study[42] (Exam 8; $n = 2724$) were stratified into high DMI and low DMI groups based on the cohort-wide median. This categorical variable was used to perform Kaplan–Meier survival analysis using the lifelines Python package[69]. Cox proportional hazards (CoxPH) models incorporated DMI, age, and sex as covariates, with further adjustment for 12 distinct immune cell types[43]. Outcomes analyzed were overall mortality, cardiovascular disease, coronary heart disease, and congestive heart failure. Only individuals without prior occurrence of these endpoints at the time of sampling were included ($n = 2281$). Confidence intervals (95%) were computed using the log-log method. The same methodology was applied in the cardiogenic shock sub-cohort to assess the association between DMI and survival among patients already diagnosed with the disease. In multivariate CoxPH models, DMI, age, and clonal hematopoiesis status were included as covariates. Clonal hematopoiesis was treated as a categorical variable, distinguishing individuals harboring somatic mutations in *DNMT3A* or *TET2* from those without mutations.

## Genomic distribution of ESLs

The association of ESLs with CpG islands, shores, shelves, and open sea regions was extracted directly from the Illumina 450 K bead-array manifest. Enrichment analysis for ESLs within these regional categories was performed using Fisher's exact test. Additionally, genomic coordinates of TSS were obtained from the UCSC Genome Browser annotation track database (https://genome.ucsc.edu/cgi-bin/hgTables; specifically, "NCBI RefSeq" track from the "Genes and Gene Predictions" group). Promoter regions were defined as the area from 1500 bp upstream to 1500 bp downstream of a TSS. The enrichment of ESLs within these regions was assessed using a two-sided Fisher's exact test, with non-stable CpG loci ($n = 359,459$) serving as the reference set.

## Identification of ESL-regulated, age-associated genes through integrated methylation, expression, and recurrence analysis

To identify genes for which ESL perturbation significantly influences their expression, we conducted a pan-cancer analysis using harmonized DNA methylation and RNA-seq transcript per million (TPM) data from 8960 solid cancer samples (TCGA Pan-Cancer dataset downloaded from https://xenabrowser.net/datapages/). We focused on recurrently perturbed ESLs located within 1500 bp upstream of TSSs, resulting in 3,584 sites associated with 1916 genes after intersecting with the TCGA data. For each ESL, Pearson's correlation coefficient was calculated between the methylation level (β-value) and the expression (TPM) of the corresponding gene. For genes with multiple ESLs in their promoter regions, the mean correlation coefficient (r) was calculated. In a second analysis, the correlation between gene expression and age was calculated for each of the 1916 genes using a dataset of healthy individuals from the Genotype-Tissue Expression (GTEx) project[45] who had their peripheral blood assayed with RNA-seq. In a third analysis, we calculated the mean recurrence across ESLs within each gene promoter. These steps yielded three variables for each gene: methylation versus expression (*r*), age versus expression (*r*), and recurrence (Supplementary Data 5). A total of 236 genes were selected for further interrogation based on the following criteria: (1) a significant negative correlation between age and expression, (2) a significant negative correlation between methylation at promoter-related ESLs and expression, and (3) a mean ESL recurrence greater than 5%. Statistical significance was determined using an alpha of 0.05, with Bonferroni correction applied for multiple hypothesis testing.

## Differential methylation analysis

CpG islands located in the promoter regions of the 236 age-dependent genes were analyzed using whole-genome bisulfite sequencing data from six genetically diverse AML samples (IDs: 412761, 868442, 400220, 545259, 548327, 573988) sourced from Wilson et al.[70], along with six CD34+ cells from healthy donors as controls. Differential methylation between the AML and control samples was assessed by comparing beta values for each CpG island and its associated ESLs using a two-sided Wilcoxon rank-sum test. A *P*-value < 0.05, adjusted for multiple comparisons using the Bonferroni correction, was considered statistically significant.

## Transcription factor motif mapping

To investigate the regulatory mechanisms underlying age-dependent gene expression changes, gene set enrichment analysis was performed using g:Profiler[71] on a curated list of 236 age-associated genes identified through the analysis described above. To evaluate whether TF binding sites overlapped with ESLs, we utilized JASPAR[72] motif coordinates through the "JASPAR Transcription Factors" track available via the UCSC Genome Browser. Only motifs with JASPAR position weight matrix scores corresponding to a *P*-value below 0.0001 (i.e., scores greater than 400) were retained. Intersection of these filtered motif sites with the genomic coordinates of ESLs resulted in TF binding motifs containing ESLs across age-related genes.

## Permutation tests

For the analysis pairing diagnosis and relapse samples based on low-recurrence, highly destabilized ESLs, a permutation test was performed to evaluate the probability of obtaining a similar matching success rate by chance. For each cancer-type dataset, 1,000 simulations were performed in which 15 random low-recurrence ESLs were selected for each patient. In each iteration, the total number of randomly selected sites ($n = 15$ times the number of patients) was then used to perform unsupervised clustering, with Ward's linkage method and Pearson's correlation as the distance metric. The *P*-value was defined as the proportion of simulations in which the number of correctly paired patients equaled or exceeded the observed count, reflecting the likelihood of achieving such a result by chance.

## Statistics and reproducibility

Statistical analysis was done with R/RStudio version 4.2.2 and Python version 3.1.1. Statistics for each analysis are described in the relevant section. For the cardiogenic shock cohort, sample size was determined by power analysis. For all other cohorts, sample sizes were based on the maximum number of available samples or data. In the Framingham Heart Study cohort, participants who had already experienced the endpoints of interest prior to sampling were excluded; no other samples were excluded. As this study retrospectively examines associations between abnormal methylation at specific loci and disease states, no treatments were tested, and the experiments were not randomized. The Investigators were not blinded to allocation during experiments and outcome assessment, as blinding was not relevant for the reported analyses.

## Consideration of sex and gender

CpG sites on sex chromosomes were excluded to ensure results are generalizable across sexes. Perturbation of autosomal ESLs was analyzed separately in males and females with chronic lymphocytic leukemia, demonstrating that this phenomenon occurs independently of sex. Biological sex was included as a covariate in Cox proportional hazards regression analyses of the Framingham Heart Study cohort. Gender-based analyses were not carried out due to the limited reporting of gender information in the available datasets.

## Reporting summary

Further information on research design is available in the Nature Portfolio Reporting Summary linked to this article.

## Data availability

The sources of publicly available datasets used in this study are listed in Supplementary Data 1. Newly generated methylation data of longitudinal AML samples, cardiogenic shock patients, MDS patients, and CML patients have been deposited in the GEO database under accession numbers GSE315367, GSE315366, and GSE315451. Source data are provided with this paper.

## Code availability

All scripts used to perform data analysis and produce figures are available at https://github.com/abelson-lab/DNA-Methylation-Instability.

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

## Acknowledgements

We sincerely acknowledge and thank all the patients who contributed biological samples, as well as the studies that made their data publicly available to support this research. We are also grateful to the Centre for Applied Genomics at The Hospital for Sick Children in Toronto, Canada, for their assistance with DNAm profiling. Our deepest thanks go to John E. Dick for his invaluable support in the cardiogenic shock and longitudinal AML studies, which provided critical datasets for this work, as well as for his review of the manuscript. We also thank Robert J. Vanner and Alex Murison for their thoughtful and critical review of the manuscript. This research was funded by an Investigator Award from the Ontario Institute for Cancer Research, supported by the province of Ontario (to S.A.). Additional funding was provided by the Canada Graduate Scholarship from the Canadian Institutes of Health Research and the Sona Naran Pancha Graduate Award in Leukemia Research (to S.B.).

## Author contributions

S.B. performed all data analyses, interpreted results, contributed to concept development, and co-wrote the manuscript. I.N.M. contributed to single-cell data analysis and sample processing. R.D., F.L.S., M.B., A.A., J.A., and S.V.B. facilitated sample and data acquisition and clinical data curation. T.M., S.M.C., M.D.M., D.D.H.K., R.K., and F.B. managed patient recruitment, data, and sample acquisition and provided clinical expertise. S.A. conceptualized and designed the study, contributed to data analysis, led and supervised all aspects of the study, and co-wrote the manuscript. All authors read and approved the manuscript.

## Competing interests

S.V.B. is a co-inventor on patents related to mutation and methylation analysis in cell-free DNA that have been licensed to Roche and Adela, respectively, and he is co-founder and has ownership in Adela. The remaining authors declare no competing interests.
