## [Transparent Peer Review file · Nature Communications]

Blood-Based Epigenetic Instability Linked to Human Aging and Disease

Corresponding Author: Dr Sagi Abelson

Version 0:

Reviewer comments:

Reviewer #1

(Remarks to the Author)

This manuscript entitled “Blood-Based Epigenetic Instability Linked to Human Aging and Disease” by Salman Basrai and colleagues presents a new computational framework to define and identify blood-based Epigenetically Stable Loci (ESLs). The authors characterize the frequency, abundance, and variability of these ESLs in blood malignancies, investigate their perturbation in relation to chromatin accessibility and lineage specificity, and explore their potential role on disease states such as in leukemia diagnosis, remission, and relapse. The study further reveals associations between ESL perturbations and known CHIP driver mutations, and how these may contribute to cardiovascular risk and mortality. Additionally, evidence suggests progressive epigenetic instability with aging and demonstrates potential regulatory mechanisms for disease-associated genes through ESL perturbation.

The new computational approach used for identification of ESLs-CpG sites with consist high and low methylation level, relied on approximately 1,700 young healthy adult DNA methylation profiles. Although these ESLs identified from blood, their methylation pattern is conserved across 82 cell types. Yet, their methylation level exhibits variability among blood malignancies, where such variability varies among different types of blood cancer. Single-cell ATAC-seq profiles of healthy hematopoietic cells uncover that lymphoid cancer-enriched ESLs, which are easier to be hypermethylated, exhibited low accessibility in common lymphoid progenitors. Conversely, myeloid-enriched perturbed ESLs exhibited significantly reduced chromatin accessibility in HSCs and myeloid progenitors. These results support a model in which ESL instability could arise as early events, occurring prior to or alongside the initiation of malignancy. Further, analysis of somatic-perturbed ESLs in a cohort of 3,019 cancer samples demonstrates that these sporadic epigenetic events correlate with disease diagnosis, remission, and relapse. To quantify this, the study introduces DNA Methylation Instability (DMI) as a per-sample measure of ESL perturbation. Importantly, DMI also increases progressively with age in healthy individuals, indicating that epigenetic instability is one of features of aging hematopoietic cells. Elevated DMI is further linked to increased risk of cardiovascular disease and mortality, independently of clonal hematopoiesis. Mechanistically, ESLs are enriched around transcription start sites and overlap with key transcription factor motifs; their perturbation is associated with downregulation of age- and disease-associated genes, such as PRDM5 and RGS2. Together, these findings support a model in which ESL instability contributes to aging and disease by disrupting transcriptional regulation and enabling clonal expansion in hematopoiesis.

The findings of this study are important to advance our understanding of how epigenetic variation can drive altered cellular behaviors in human health and disease. This study proposes a new concept Epigenetically Stable Loci (ESLs) as a reference to quantify epigenetic state deviation across diseases and aging.

While this is a valuable paper in my opinion, I have a few suggestions for improving this work and contextualizing the findings. Most importantly, it would be helpful for the authors to clearly discuss how DNA methylation instability could arise either from alterations in clonality (e.g. a clone with a specific epigenetic variant could emerge and dominate) or from epigenome alterations within a clone. In the former case, the marks might only track clones, but be innocuous, while in the latter case, the marks might alter clonal behavior. It would be good to discuss this and also put this into the context of how other markers such as nuclear somatic mutations (doi: 10.1038/s41586-022-04786-y), mtDNA mutations (doi: 10.1038/s41586-024-07066-z), or even epi-mutations themselves (dois: 10.1038/s41592-024-02567-1, 10.1038/s41586-025-09041-8) show variable clonal contributions with aging and disease. Importantly and in a related manner, cause-and-effect implications are often prematurely drawn in the absence of the required data, analyses, and functional validation. I outline these concerns below in detail:

Major Comments:

- Line 175: The statement “This consistency across diverse tissue types suggests the presence of stringent regulatory control established early in embryonic development...” may be overstated. First, the conclusion lacks direct evidence from embryonic tissues to support the claim of early developmental establishment. Moreover, if applying the same threshold, which was used in Fig. 2a for definition variable ESLs in blood malignancy, to different cell type in Fig. 1c, it suggests these ESLs regions still exhibit variability across normal tissues - albeit to a lesser extent than in blood cancers. Hence, it would be better to modify the conclusion here for accuracy.

- Line 212: The statement “These results support a model in which ESL instability could arise as early events, occurring prior to or alongside the initiation of malignancy” contains a logical gap. ESLs are defined based on data from healthy individuals, whereas ESL perturbation is assessed in blood cancer samples. Therefore, the analysis does not clearly distinguish whether the observed instability is a consequence of malignant transformation or simply reflects normal variation during hematopoietic development (or a process like selection of specific clones harboring drivers). This raises concerns about attributing ESL perturbation to the initiation of malignancy. In particular, the cellular composition of normal blood differs markedly from that of AML, which is characterized by an overproduction of blast cells. It is thus possible that the perturbed ESL regions identified in AML reflect regulatory loci specific to hematopoietic stem and progenitor cells, rather than loci altered because of disease progression or that this might reflect clonal dynamics. To address this, it may be helpful to separately analyze disease-perturbed ESLs and cell type-specific ESLs. At a minimum, this discussion should be modified.

- Line 222: the conclusion that “ESL instability could arise as early events, occurring prior to or alongside the initiation of malignancy” appears to overstate the implications of the correlative data and to imply a more causal relationship than is supported. The logic from ESL perturbation and chromatin accessibility to leukemogenesis remains speculative in the absence of temporal or functional evidence. Language should be adjusted accordingly. Particularly, since as noted above, this might reflect shifts in clonal contributions.

- Line 229: It is not clear what is meant by “The low statistical probability of these results occurring by chance...”

- Line 253: It may be important to know whether the “DMI regions” identified here are located close known driver mutations shown in Figure 4G. Are these regions proximal to cancer driver genes or potentially involved in regulating their expression? A systematic analysis or annotation of nearby genes and their known disease relevance would be valuable for interpretation. In this case, this could provide a mechanistic link between somatic epigenetic alterations and genetic drivers of leukemogenesis.

- Line 270–272: The conclusion that “methylation instability accelerates following malignant transformation” may be overstated, as it is based on comparisons involving only eight leukemia patients (in main) and six patients (in Supp). Such a small sample size limits the generalizability of the findings.

- Line 295: Are these DMI regions are close to known CHIP driver mutations?

- Line 331: The current evidence illustrating the connection between RGS2 ESL and cardiovascular disease remains correlative. The phrasing “may play a mechanistic role” implies a causal relationship, which is not directly demonstrated in the data presented.

- Line 334: There’s no direct evidence to connect ESL and clonal expansion.

- Line 348: This statement “methylation patterns are established during early development...” is not supported by data presented in this manuscript.

- Line 358: I recommend the term “detectable mutations” is modified to :known driver mutations” for accuracy

Minor Comments:

- Fig. 1b. This figure should clearly introduce the term “ESLs,” referring to the stable unmethylated and stable methylated loci. Currently, this definition is only described in the main text, which may cause confusion, particularly since this term “ESL” is introduced in the legend of Fig. 1c without being properly highlighted earlier. It would be helpful to explicitly present this concept in Fig. 1b. Additionally, the number of stable unmethylated and stable methylated loci should be indicated in the figure to provide quantitative context.

- Fig. 1c: The legend states, “Violin plots showing β -values of methylated (top) and unmethylated (bottom) ESLs across 82 cell types.” However, it should be clarified that the ESLs plotted here are those identified from the cohort of 1,658 young adults shown in Figure 1a, rather than ESLs independently defined within each cell type or tissue. This clarification is important to avoid the misunderstanding that ESLs were separately called in each dataset.

- Line 189: There appears to be a mistake in the statement: “Hypomethylation of normally methylated ESLs was also observed... in Supplemental Figure 2.” Based on Supplemental Figure 2, the pattern reflects “hypermethylation” of normally methylated ESLs, not “hypomethylation”.

- Line 196: It would be helpful to define “perturbed ESLs” in the main text, not only in the Methods section, since subsequent analyses heavily rely on this concept.

(Remarks on code availability)
Appropriate code provided

Reviewer #2

(Remarks to the Author)
I co-reviewed this manuscript with one of the reviewers who provided the listed reports. This is part of the Nature Communications initiative to facilitate training in peer review and to provide appropriate recognition for Early Career Researchers who co-review manuscripts.

(Remarks on code availability)

Reviewer #3

(Remarks to the Author)
The paper “Blood-Based Epigenetic Instability Linked to Human Aging and Disease” by Basrai S et al studies the biological and clinical significance of a particular age-related DNA methylation pattern, characterized by loci (termed epigenetic stable loci-ESLs) that are stably unmethylated or methylated across over 1000 blood samples from young individuals (all aged 18), but which are demonstrated to undergo destabilization in hematological cancers, and which in other cohorts (e.g. the FHS) is informative of cardiovascular-disease and mortality. An important emerging insight from this study is that the DNA methylation instability (quantified through a measure called DMI) is associated with cardiovascular disease and mortality risk in a manner that seems to be independent of clonal hematopoiesis (CH). High DMI values are also observed in individuals with no clear genetically driven CH, pointing towards other potential sources of CH.
Overall, I think that this is an interesting study, although I do unfortunately have some major serious concerns, some regarding statements about true conceptual novelty, and others of a more technical nature. I now describe these in more detail:

Major concerns:

1) Conceptual novelty is questionable: The concept to focus on epigenetically stable loci in a normal reference population, and to then study whether these loci can be informative of disease risk is not entirely novel, as it has been explored in various different contexts. For instance, a 2012 study (Teschendorff et al Genome Med 2012) used differential variance as a means of feature selection to demonstrate that increased DNAm variability can be informative of cervical cancer risk, 3 years in advance of diagnosis. Whilst, the tissue-type and context differs, that approach, termed EVORA, selected for CpGs that are ultra-stable across the normal samples that remained healthy. Other studies have explored the role of stochastic epimutations in blood in the context of large cohorts such as EPIC-Italy, and I can refer the authors to this recent review (Teschendorff & Horvath Nat Rev Genet 2025) where some of these studies were cited. Hence, I would recommend that the authors rewrite the Introduction to more clearly compare the approach taken here to those of previous epigenetic studies, as described above.

2) Technical concerns when identifying ESLs: (a) It is a well-known fact that the top PCs of any WB DNAm dataset would correlate very strongly with variations in immune-cell fractions, notably with variations in the neutrophil fraction-the most abundant fraction in WB. Thus, when you search for ESL in WB the most sensible approach would be to focus on loci that are not immune-cell type dependent, say loci that are stably unmethylated <0.2 across all immune cell-types. For these loci, how many neutrophils or lymphocytes there are in any given sample is completely irrelevant, because all underlying cell-types are in the same unmethylated state. Hence, the rationale of the authors to remove samples that display abnormally high or low levels of any given immune cell-type is not well justified. It is completely unnecessary. (b) The authors state that “Probes with detection $P > 1 \times 10^{-10}$ in more than 5% of samples were removed using the detection function”. This super stringent threshold is insane and completely unjustified. Normally, we would trust DNAm-values that pass a detection P-value threshold < 0.001 . So let us say that a CpG has a detection P-value of $1e^{-9}$ in 10% of the samples and a detection P-value of $1e^{-11}$ in the rest. It is in my opinion not well justified to remove it!!!! It is bad statistical practice and it would be damaging to the community if such errors make their way into published literature. The authors need to redo this. (c) It is also very unconventional and a bad idea to divide CpGs up into only 2 states: methylated ($\beta > 0.5$) and unmethylated (< 0.5). In any given cell, DNAm can be in 3 states, and so when we talk about a CpG being unmethylated, one should adopt a much more stringent threshold, for instance $\beta < 0.2$. Likewise, fully methylated would be $\beta > 0.75$ (there is no direct symmetry due to technical assay consideration, hence why the threshold is lowered slightly), with the rest being in a partially or semi-methylated state ($0.2 < \beta < 0.75$). Moreover, later on line 170-172 the authors state that no ESLs displayed intermediate values, but according to the definition given in Methods, there could never be an intermediate state, since the authors used 0.5 as a threshold to call CpGs unmethylated or methylated. It just does not make sense!

3) ESLs are stable but not ultra-stable: When I inspect Fig.1C, I don't appreciate that these loci are that stable. Whilst the bulk of the beta-values for unmethylated CpGs are indeed < 0.25 , I have seen from my own experience, CpGs displaying tighter distributions. Moreover, the violins do extend across the whole range (0 to 1), and it is unclear to me whether this is just an artefact of the plotting routine, or whether indeed there are values in the whole range. Could the authors please clarify this point? For instance, the first blue violin for adipocytes, the density extends from 0 to 1, so does that mean that there are values close to 1 for this otherwise unmethylated ESL? If so, what is the interpretation of these outliers. I note that the same patterns are observed in blood cell-types, which is more relevant.

4) Biological significance of Fig.3 is unclear: I would like to better understand the biological significance of this figure. If I

understood correctly, ESLs that are more perturbed in myeloid malignancies are less accessible in normal myeloid cells, whereas ESLs more perturbed in lymphoid malignancies are less accessible in normal lymphoid cells. So, could the authors go beyond scATAC-Seq, for instance test for chromatin-state enrichment using data from the IHEC/Epigenomic roadmap, as it suggests to me that these ESLs map to repressed regions. If so, what is the biological meaning of increased DNAm at already repressed regions? That most of it is non-functional?

5) Fig.4A seems trivial to me: perhaps I misunderstood, because it is unclear what is being displayed here. But if the x-axis measures for a given ESL, the fraction of samples that are perturbed at this locus, then clearly the average beta-value of that locus over samples will increase since the ESLs are all unmethylated in the normal reference state. Hence, isn't the correlation being depicted in Fig.4A trivial? I don't see what it shows.

6) SuppFig8: Related to point-4 above, in this figure the authors display around 234 genes whose promoters contain ESLs and which display downregulation with age. However, the original number of genes with ESLs was 1,879 genes, so my question is what happened to the rest of genes (n= 1645)? Did they display overexpression with age or no change? I think this correlative analysis with gene-expression should be expanded: for instance what happens to ESLs that are actually methylated? Do they map to promoters or other gene regions and does their expression change with age? if the claim is that these age-associated DNAm changes at ESLs are consistently associated with changes in gene-expression, that we need a formal global statistical analysis with P-values confirming such a claim.

7) Significance of motif enrichment is unclear (Fig.7): I also found the results of Fig.7 unclear in relation to what it all means. I understand that increased DNAm at these sites may effect TF-binding, and the authors identify a number of TFs for which binding motifs are enriched among the ESLs. But there is no clear validation of this in relation to any of the identified TFs. Some form of in-silico validation could have attempted. Moreover, I find panel Fig.7C in particular (and to some degree also Fig.7D) to be very unclear. Why do the authors revert to TCGA data? What TCGA cancer-types were included? Did this analysis adjust for cell-type heterogeneity? If anything, panel 7C seems to contradict that authors claim, because I see more datapoints on the positive side of the y-axis indicating a preponderance of positive correlations between promoter DNAm and gene expression?

8) Adjustment for cell-type heterogeneity when correlating to health outcomes: Related to some of the earlier points, the loci the authors call ESL still display some variation, say on the scale of 0-25% changes in DNAm. I worry that this variation could be driven by unaccounted cell-type heterogeneity, for instance, shifts between the memory and naïve T-cell fractions. Likewise, CHIP has been associated with shifts in immune-cell fractions. Hence, I think it would be extremely important if the authors could demonstrate that DMI correlates with all the health outcomes in a manner that is independent of variations in immune-cell fractions using the higher 12 IC-type DNAm reference panel in the EpiDISH Bioconductor package (see e.g. Luo Q et al Genome Med 2023), which includes memory and naïve T-cell subsets. Indeed, I note that the authors seems unaware of a fairly extensive body of literature demonstrating that naïve T-cell fractions are generally protective of mortality and CVD etc, see e.g. Luo Q et al Genome Med 2023.

(Remarks on code availability)

Reviewer #4

(Remarks to the Author)

In the manuscript entitled "Blood-Based Epigenetic Instability Linked to Human Aging and Disease" (Manuscript ID: NCOMMS-25-46973-T), Basrai et al. analyzed multiple, diverse cohorts utilizing DNA methylation data generated from the two most recent Illumina platforms: Infinium HumanMethylation450 (450k) and Infinium MethylationEPIC (EPIC) BeadChip arrays. By focusing on probes common to both platforms, the authors define a subset of CpG sites that are typically stable (either consistently methylated or unmethylated) in human peripheral blood, and across multiple tissues, so-called Epigenetically Stable Loci (ESLs). Subsequent analysis demonstrated that these ESLs become epigenetically unstable in hematological malignancies showing aberrant methylation levels. Furthermore, the authors observed that these loci increase variability in methylation with age, and this variability is associated with an elevated risk of developing cardiovascular disease. Based on these observations, the authors developed a new metric called "DNA methylation instability" (DMI), which quantifies the degree of variability at these previously stable CpG sites (especially those that acquire abnormal methylation). They demonstrated that this metric strongly correlates with biological aging and disease progression, suggesting a potential utility for monitoring disease risk or state. Overall, I found the manuscript an interesting paper to read, and the proposed DMI metric holds potential translational interest as it correlates with leukemic burden, aging and cardiovascular risk. While the integration of large-scale methylation datasets, single-cell chromatin accessibility, and multiple cohorts enhances the robustness of the findings, I have raised several issues that should be addressed before this manuscript can be considered for publication. My major concerns are outlined below, followed by several minor ones:

Major points

(1) Given that the study involves multiple datasets generated from two different platforms and distinct cohorts, with a focus on methylation, the authors should provide additional details on the pre-processing of the DNA methylation data. For instance, it is unclear whether the arrays from different cohorts were processed together as a single dataset or separately on a cohort-wise basis. Clarifying the pre-processing workflow, including steps such as quality control, normalization methods or batch effect correction, would strengthen both the rigor and reproducibility of the study. Incorporating exploratory analyses, such as principal component analysis (PCA), is also recommended to assess data concordance across cohorts and platforms. Additionally, since the analysis was limited to CpG sites shared between the 450k and EPIC arrays, the authors should demonstrate that beta (β) values are consistent across platforms. Differences in probe chemistry or hybridization efficiency between the 450K and EPIC arrays could introduce technical biases that might confound results. A validation step, such as comparing β -values at overlapping CpG sites in a subset of samples profiled on both platforms, would help confirm cross-platform compatibility and therefore, robustness of the reported findings.

(2) The authors report a significant enrichment of ESLs for CpG islands (OR = ?; $p < 0.001$). However, this observation could be influenced by the inherent design of the Infinium MethylationEPIC/450k array, which is biased for CpG-rich regions such as gene promoters and CpG islands (CpGi) as acknowledged in the discussion. As a result, the apparent enrichment for CpGi could reflect the array probe distribution rather than a true biological signal. To strengthen their conclusions, the authors should clarify whether they accounted for this platform-specific bias in their analysis. If not, the observed enrichment may be an artifact of array design. Normalizing their findings against the EPIC array's probe distribution or explicitly addressing further this confounder in the discussion would improve the robustness of their interpretation. The authors may also consider including values corresponding to fold enrichment in the main text.

(3) The authors reported a number of stable CpG sites. However, to provide greater biological insight, it would be important to determine whether these CpGs represent isolated events or whether they cluster within a broader differentially methylated region (DMRs). Regional methylation changes often have stronger functional relevance than single-site alterations. Therefore, the authors should investigate whether these CpGs cluster within DMRs. If DMRs are found, the authors could include illustrative plots of methylation profiles across representative DMRs, which would help to demonstrate the regional patterns of methylation stability observed in the study.

(4) The authors validate the consistency of DNA methylation levels at the identified ESLs using whole-genome bisulfite sequencing data from 82 human cell types. To further support the epigenetic stability of these loci, the authors could also examine methylation levels at these ESLs in GTEx samples (PMID: 36510025). This additional analysis would provide complementary evidence of ESL stability across a range of non-hematologic tissues, further reinforcing their findings.

(5) The authors should provide more comprehensive data. Currently, the supplementary tables only list Probe IDs for the stable CpG sites (and coordinates, in some cases based on the hg19 assembly). This should be supplemented with summary statistics describing the methylation profiles observed across the samples used to establish the thresholds. In addition, the data should include information on the genomic context (e.g., promoter, exon, intron, intergenic) and closest gene to each CpG site as well as, if not present, the genomic coordinates ideally referenced to the hg38 genome build, as this is the most widely used reference genome today. These details should be accompanied by plots illustrating methylation changes, as the current materials do not show the magnitude of changes observed in samples with epigenetic alterations. Including these elements would substantially improve data transparency and interpretability.

(6) The thresholds established to define "perturbed" ESLs were established based on β -values at these CpG sites across three independent cohorts of healthy individuals: GSE124413, GSE132203 and GSE141682. The authors should clearly justify why they used these three particular cohorts for setting such a threshold (99.9th percentile of pooled β -values across ESLs). Currently, these three cohorts are only briefly described in the supplemental Table 1. Potential confounders (e.g., age, sex and ethnicity) could influence baseline methylation levels.

7) The criteria for defining ESLs appear somewhat arbitrary and lack thorough justification (10% least variable sites across the discovery cohort). A more detailed rationale for this threshold would improve confidence in the robustness of the definition. This concern also applies to the criteria used to identify lineage-specific perturbed ESLs, which similarly appear to be selected without clear justification.

8) In the survival analysis, participants from the Framingham Heart Study were stratified into "high DMI" and "low DMI" groups based on the cohort-wide median. However, dichotomizing a continuous variable like DMI can reduce within-group variance and may limit the detection of more specific associations with outcomes. As a complement to the dichotomized analysis, the authors could present the full distribution of DMI across participants, either as a supplemental figure or table, to provide a clearer picture of the range and variability in the cohort. Additionally, it would be valuable to repeat the analysis focusing on individuals with extreme DMI values (for example, those below the 10th percentile and above the 90th percentile) to assess whether associations are stronger or more pronounced at the distribution's tails. This approach could help evaluate the robustness and sensitivity of the findings.

9) Following the analysis of ESL methylation versus gene expression and age versus gene expression (and recurrence), 234 genes were selected for further investigation. However, the criteria for determining significance, including whether any multiple testing correction was applied, are not clearly described. The authors should provide these details.

10) It is unclear whether the data from the "Differential methylation analysis" included in the supplementary material are provided as a supplementary table. Including such a table could help illustrate the epigenetic changes occurring across these regions. Additionally, Supplementary Figure 9 can be supplemented with further examples (please add methylation scale on y-axis).

Minor points.

11) Although the study delivers valuable insights into the stability and perturbation of the human methylome across health, disease, and aging, the current title may slightly overstate its scope. Specifically, the title implies a broad investigation of disease-related epigenetic changes, whereas the study primarily focuses on two disease categories: cardiovascular disease and hematopoietic malignancies. To better reflect the manuscript's content, consider narrowing the title to align with the studied conditions.

12) The abstract may benefit from clearly presenting the study's scale through the inclusion of key numbers, such as the exact number of CpG sites found to have consistently stable methylation profiles in healthy blood, the statistical significance thresholds or criteria used for site selection or total number of cohorts analyzed, along with sample size for each cohorts (for

example: cardiovascular disease cohort, n = X,XXX; healthy control cohort, n = X,XXX). Presenting these specific numbers would help readers better appreciate the scale and robustness of the work. As currently written, the abstract lacks these critical metrics, making it feel somewhat vague.

13) The introduction feels rather brief and concludes abruptly after introducing DMI (Line 80). Adding a paragraph introducing clearly the central research question, the study's hypothesis, or specific objectives, along with a brief description on how the study addresses this question would strengthen this section. Additionally, while the manuscript focus is on epigenetic mechanisms, the introduction focuses predominantly on genetic factors, providing limited discussion of epigenetic dysregulation in aging or clonal hematopoiesis (CH). Expanding the introduction to include a concise overview of previous findings on methylation instability in CH, with references to relevant studies, would offer valuable context and underscore the novelty and importance of the current work. This added background will equip readers with a stronger foundation for understanding the motivation behind the study and allow for a clearer appreciation of the subsequent sections.

14) Figure 2B. A more informative representation would be a violin plot combined with an overlaid boxplot. This visualization would provide a more complete view of the data by showing the entire distribution shape plus key summary statistics such as median, quartiles, etc. This combined approach provides much more insight than the current presentation of only the median for each disorder.

15) Figure 5. I would re-labeled x-axis to indicate intervals.

16) Figure 6D. I would recommend removing the table placed to the right of the plot and increasing the plot size for better visualization. The p-values could either be directly incorporated into the plot or color-coded based on significance. The table content could then be moved to the main text or included as a supplementary table.

17) Line 138-139. The authors stated: Clonal hematopoiesis was treated as a binary variable indicating the presence or absence of mutations. Do the authors refer to mutations at specific sets of genes? If so, the authors should mention some genes or cite relevant bibliography.

(Remarks on code availability)

Version 1:

Reviewer comments:

Reviewer #1

(Remarks to the Author)

I appreciate the thorough and thoughtful revisions. The point-by-point responses were very clear and the additional analyses, clarifications, and textual modifications appropriately addressed all major concerns raised in the initial review. In particular, the authors have:

- (1) Clarified differences between clonal alterations vs. epigenomic changes in the discussion.
- (2) Added new developmental methylation data and revised interpretations related to early events in malignancy to avoid overstating implications.
- (3) Strengthened statistical methods for clonal preservation analyses.
- (4) Provided helpful annotations linking ESL-associated genes to diseases.
- (5) Adjusted figure legends and terminology for improved clarity and reproducibility, and made numerous careful textual improvements throughout the manuscript.

Overall, the authors have resolved the major conceptual and technical concerns from the initial review. I have no additional concerns and believe this work is appropriate for publication.

(Remarks on code availability)

Reviewer #2

(Remarks to the Author)

(Remarks on code availability)

Reviewer #3

(Remarks to the Author)

I thank the authors for thoroughly addressing the concerns I had. I think they did a fantastic job and MS is much improved. There is however just one remaining issue regarding some missing and a wrong citation that should be addressed:

Major points:

1-In the Introduction, when referring to studies that have explored DNAm outliers, there are other blood-based studies that

precede Conboy's work, and which have not been cited. For instance, Gentilini D et al Aging 2015 <https://doi.org/10.18632/aging.100792> , Gentilini D et al J Endocrinol Invest DOI: 10.1007/s40618-022-01915-2 .

2- The study of Meyer and Schumacher (Ref.25) has absolutely nothing to do with DNAm outliers, so it is incorrect and misleading to cite this work in this context. Please remove it, because stochasticity and outliers are 2 different concepts and it is wrong to confuse the two: outliers could be stochastic (as for instance shown in Teschendorff et al Nat Commun 2016) but they don't have to be, and conversely, many non-outliers can display stochasticity (e.g. there is evidence that the Zhang clock CpGs could be changing stochastically at the single-cell level, Tong H et al Nat Aging 2024) and more generally Tarkhov AE et al Nat Aging 2024).

Minor (optional): It should also be noted that historically, biological and clinical relevance of DNAm outliers were first studied in non-blood tissues and that the blood-based studies came later. I am not sure if the relevant sentences in this paragraph should be reordered to emphasize the historical timeline.

(Remarks on code availability)

Reviewer #4

(Remarks to the Author)

The authors have provided thorough, point-by-point responses to all of my concerns and have revised the manuscript accordingly. The additional analyses, new figures, and expanded introduction and discussion sections strengthen the manuscript and fully address the issues raised. I therefore recommend this manuscript for acceptance and congratulate the authors on their excellent work.

I have just a couple of very minor comments:

1. In the introduction, both "CpG sites" and "CpG loci" are used. For consistency, I suggest selecting one term and employing it throughout the manuscript.
2. The acronym "CpGs" has not been defined. Similarly, the acronym for clonal hematopoiesis (CH) is defined in the Discussion despite this term being used earlier in the text. The authors should consider defining these acronyms when their corresponding terms are first introduced.
3. In the methods section, the authors stated that "We also excluded non-CpG (CpH) targeting probes and probes overlapping SNPs." It would be helpful to clarify whether this exclusion applies to all SNPs or specifically to common SNPs (e.g., those with a minor allele frequency > 0.05).

(Remarks on code availability)

We would like to thank the reviewers for their time, thoughtful evaluation of our manuscript, and for the insightful comments and suggestions that have helped strengthen our work. We have carefully considered all feedback and revised the manuscript accordingly. Below, we provide detailed responses to each comment and indicate the corresponding changes made in the paper.

Reviewer #1 (Remarks to the Author):

This manuscript entitled "Blood-Based Epigenetic Instability Linked to Human Aging and Disease" by Salman Basrai and colleagues presents a new computational framework to define and identify blood-based Epigenetically Stable Loci (ESLs). The authors characterize the frequency, abundance, and variability of these ESLs in blood malignancies, investigate their perturbation in relation to chromatin accessibility and lineage specificity, and explore their potential role on disease states such as in leukemia diagnosis, remission, and relapse. The study further reveals associations between ESL perturbations and known CHIP driver mutations, and how these may contribute to cardiovascular risk and mortality. Additionally, evidence suggests progressive epigenetic instability with aging and demonstrates potential regulatory mechanisms for disease-associated genes through ESL perturbation.

The new computational approach used for identification of ESLs-CpG sites with consist high and low methylation level, relied on approximately 1,700 young healthy adult DNA methylation profiles. Although these ESLs identified from blood, their methylation pattern is conserved across 82 cell types. Yet, their methylation level exhibits variability among blood malignancies, where such variability varies among different types of blood cancer. Single-cell ATAC-seq profiles of healthy hematopoietic cells uncover that lymphoid cancer-enriched ESLs, which are easier to be hypermethylated, exhibited low accessibility in common lymphoid progenitors. Conversely, myeloid-enriched perturbed ESLs exhibited significantly reduced chromatin accessibility in HSCs and myeloid progenitors. These results support a model in which ESL instability could arise as early events, occurring prior to or alongside the initiation of malignancy. Further, analysis of somatic-perturbed ESLs in a cohort of 3,019 cancer samples demonstrates that these sporadic epigenetic events correlate with disease diagnosis, remission, and relapse. To quantify this, the study introduces DNA Methylation Instability (DMI) as a per-sample measure of ESL perturbation. Importantly, DMI also increases progressively with age in healthy individuals, indicating that epigenetic instability is one of features of aging hematopoietic cells. Elevated DMI is further linked to increased risk of cardiovascular disease and mortality, independently of clonal hematopoiesis. Mechanistically, ESLs are enriched around transcription start sites and overlap with key transcription factor motifs; their perturbation is associated with downregulation of age- and disease-associated genes, such as PRDM5 and RGS2. Together, these findings support a model in which ESL instability contributes to aging and disease by disrupting transcriptional regulation and enabling clonal expansion in hematopoiesis.

The findings of this study are important to advance our understanding of how epigenetic variation can drive altered cellular behaviors in human health and disease. This study proposes a new concept Epigenetically Stable Loci (ESLs) as a reference to quantify epigenetic state deviation across diseases and aging.

While this is a valuable paper in my opinion, I have a few suggestions for improving this work and contextualizing the findings. Most importantly, it would be helpful for the authors to clearly discuss how DNA methylation instability could arise either from alterations in clonality (e.g. a clone with a specific epigenetic variant could emerge and dominate) or from epigenome alterations within a clone. In the former case, the marks might only track clones, but be innocuous, while in the latter case, the marks might alter clonal behavior. It would be good to discuss this and also put this into the context of how other markers such as nuclear somatic mutations (doi: 10.1038/s41586-022-04786-y), mtDNA mutations (doi: 10.1038/s41586-024-07066-z), or even epi-mutations themselves (dois: 10.1038/s41592-024-02567-1, 10.1038/s41586-025-09041-8) show variable clonal contributions with aging and disease. Importantly and in a related manner, cause-and-effect implications are often prematurely drawn in the absence of the required data, analyses, and functional validation. I outline these concerns below in detail:

We are grateful for the reviewer's kind recognition of the significance and value of our study.

To better communicate the implications of our work and place our findings in context, we have added a new section to the Discussion (**Lines 409–422**). In response to the reviewer's concerns regarding potential overinterpretation of causality, we now provide a detailed point-by-point response and additional analyses to clarify and contextualize our conclusions.

Major Comments:

- Line 175: The statement "This consistency across diverse tissue types suggests the presence of stringent regulatory control established early in embryonic development..." may be overstated. First, the conclusion lacks direct evidence from embryonic tissues to support the claim of early developmental establishment. Moreover, if applying the same threshold, which was used in Fig. 2a for definition variable ESLs in blood malignancy, to different cell type in Fig. 1c, it suggests these ESLs regions still exhibit variability across normal tissues - albeit to a lesser extent than in blood cancers. Hence, it would be better to modify the conclusion here for accuracy.

- Line 348: This statement "methylation patterns are established during early development..." is not supported by data presented in this manuscript.

Because the two reviewer comments above concern a related issue, we address them jointly.

We agree with the reviewer's comment regarding the need for supporting evidence for the statement, "This consistency across diverse tissue types suggests the presence of stringent regulatory control established early in embryonic development". To address this, we now include new data showing that ESLs display characteristic extreme hyper- or hypomethylation patterns already during early development (**Supplementary Fig. 2**). The corresponding section of the text has been updated to reflect this addition (**Lines 189–195**).

It now reads as follows:

"Notably, although these ESLs were identified in blood, we were able to detect their characteristic methylation patterns across non-hematological tissues using different profiling technologies (Fig. 1C). While some variation exists, the overall patterns, largely extreme hyper- or hypomethylation across ESLs, are consistently preserved across diverse tissue types (Supplementary Fig. 2A) and developmental stages (Supplementary Fig. 2B-D), suggesting the presence of stringent regulatory control established early in embryonic development and largely maintained throughout human life."

We now acknowledge that ESL regions exhibit some variability across normal tissues (**Line 191**). However, we wish to provide here comparative evidence across tissues where both array- and sequencing-based data were available, highlighting that such apparent variability should be interpreted with caution, as it may be influenced by differences in assay characteristics (**Figure R1**). Consequently, applying the array-based destabilization threshold used in Fig. 2A directly to bisulfite sequencing data would be problematic.

Figure R1. Comparison of ESL β -values measured by methylation arrays and whole-genome bisulfite sequencing. Violin plots show β -values of unmethylated ESLs ($n = 31,744$) identified in blood across other healthy tissues. For each tissue, samples assayed on the Illumina EPIC array (blue; data from Oliva et al., Nature Genetics, 2023) were compared to samples assayed by whole-genome bisulfite sequencing (orange; data from Loyfer et al., Nature, 2023). The dots indicate the proportion of β -values falling below the array-based destabilization threshold from Fig. 2A. Two y-axes are used: the left axis shows the β -value distributions, and the right axis shows the proportion of ESLs below the array-based threshold.

- Line 212: The statement “These results support a model in which ESL instability could arise as early events, occurring prior to or alongside the initiation of malignancy” contains a logical gap. ESLs are defined based on data from healthy individuals, whereas ESL perturbation is assessed in blood cancer samples. Therefore, the analysis does not clearly distinguish whether the observed instability is a consequence of malignant transformation or simply reflects normal variation during hematopoietic development (or a process like selection of specific clones harboring drivers). This raises concerns about attributing ESL perturbation to the initiation of malignancy. In particular, the cellular composition of normal blood differs markedly from that of AML, which is characterized by an overproduction of blast cells. It is thus possible that the perturbed ESL regions identified in AML reflect regulatory loci specific to hematopoietic stem and progenitor cells, rather than loci altered because of disease progression or that this might reflect clonal dynamics. To address this, it may be helpful to separately analyze disease-perturbed ESLs and cell type-specific ESLs. At a minimum, this discussion should be modified.

- Line 222: the conclusion that “ESL instability could arise as early events, occurring prior to or alongside the initiation of malignancy” appears to overstate the implications of the correlative data and to imply a more causal relationship than is supported. The logic from ESL perturbation and chromatin accessibility to leukemogenesis remains speculative in the absence of temporal or functional evidence. Language should be adjusted accordingly. Particularly, since as noted above, this might reflect shifts in clonal contributions.

Given the related content of the two reviewer comments above, it seems appropriate to address them together.

We agree with the reviewer that the referenced correlative analysis does not directly demonstrate that the observed ESL instability is a consequence of malignant transformation. At the same time, it is important to note that the perturbed myeloid-enriched ESL regions identified in AML samples are unlikely to represent regulatory loci specifically methylated in hematopoietic stem and progenitor cells (HSPCs), as ESL hypomethylation is largely consistent across different healthy blood cell types (Supplementary Fig. 1B). To directly address the reviewer’s comment regarding HSPCs, we extended this analysis in an independent dataset, focusing on the relevant myeloid-associated ESLs (**Figure R2**). These loci were found to be highly consistent, confirming that the hypermethylation observed in AML is not driven by high methylation in the regulatory landscape of normal HSPCs. To avoid implying a causal role of ESL perturbation in malignancy, we have removed the original problematic sentence (previously, line 212) and now strictly report only the observed associations.

Figure R2. Methylation level of myeloid-associated ESL across hematopoietic stem and progenitor cell types. Violin plots show the distribution of β -values for ESLs prone to hypermethylation in AML across non-malignant HSPC populations ($n = 5$ per cell type). CMP: common myeloid progenitors; GMP: granulocyte-macrophage progenitors; HSC: hematopoietic stem cells; LMPP: late multipotent progenitors; MEP: megakaryocyte-erythroid progenitors; MPP: multipotent progenitors. (Data from Jung N et al., Nat Commun. 2015)

- Line 229: It is not clear what is meant by "The low statistical probability of these results occurring by chance..."

For better clarity, we have revised the text (**Lines 249-251**). It now reads as follows:

"These results provide strong statistical evidence ($P = .002$ for AML; $P < .001$ for CLL and BCP-ALL by permutation test; Supplementary Note) for the clonal preservation of ESL methylation..."

- Line 253: It may be important to know whether the "DMI regions" identified here are located close known driver mutations shown in Figure 4G. Are these regions proximal to cancer driver genes or potentially involved in regulating their expression? A systematic analysis or annotation of nearby genes and their known disease relevance would be valuable for interpretation. In this case, this could provide a mechanistic link between somatic epigenetic alterations and genetic drivers of leukemogenesis.

- Line 295: Are these DMI regions are close to known CHIP driver mutations?

As the two reviewer comments above pertain to related points, they are addressed together for clarity and coherence.

As shown in Fig. 7A,B, DMI captures methylation changes across thousands of loci enriched in CpG islands within promoter regions and in close proximity to transcription start sites. To provide additional context for interpreting methylation in these loci, we annotated gene-disease associations for all genes containing ESLs located within or near gene bodies, untranslated regions (UTRs), or within 1,500 bp of transcription start sites, using the DISGENET database (Piñero J, et al. Nucleic Acids Res. 2017). This information has been added to **Supplementary Table 2**. Notably, among all the ESL-associated genes ($n=10,697$), 2,992 (28.0%) have been previously associated with neoplasms, and 320 of these (10.7%) had reported associations specifically with leukemias. The latter include known driver genes highlighted in Fig. 4G, Supplementary Fig. 7, and Fig. 6G (*DNMT3A* and *TET2*), associated with recurrently methylated ESLs (**Supplementary Table 2**)

- Line 270-272: The conclusion that "methylation instability accelerates following malignant transformation" may be overstated, as it is based on comparisons involving only eight leukemia patients (in main) and six patients (in Supp). Such a small sample size limits the generalizability of the findings.

We agree that the original wording may have implied reliance on only a few patients for comparison. To address this and support the robustness and generalizability of our findings, we now include a new **Supplementary Fig. 8**, which presents an analysis of DMI values across the hundreds of hematological cancer patients analyzed in Fig. 2. The text has been updated accordingly to reference this new figure (**Lines 290-292**).

- Line 331: The current evidence illustrating the connection between RGS2 ESL and cardiovascular disease remains correlative. The phrasing "may play a mechanistic role" implies a causal relationship, which is not directly demonstrated in the data presented.

We have modified the text to avoid suggesting a causal relationship (**Lines 366-368**).

It now read as follows:

"RGS2 dysregulation in blood through ESL perturbation may contribute to the observed association with cardiovascular disease, although a causal role remains to be established."

- Line 334: There's no direct evidence to connect ESL and clonal expansion.

We believe the reviewer is referring to line 344 of the original manuscript draft, which stated: "we highlight epigenetic instability as a complementary and potentially independent contributor to clonal expansion".

We have revised this sentence to more precisely reflect the finding of the study (**Lines 389-390**):
"In this study, we identify epigenetic instability as a risk factor associated with several hallmarks of clonal hematopoiesis."

- Line 358: I recommend the term "detectable mutations" is modified to "known driver mutations" for accuracy

We appreciate the reviewer's suggestion and have amended the text to use the term "known driver mutations" instead of "detectable mutations" for improved accuracy (**Line 403**).

Minor Comments:

- Fig. 1b. This figure should clearly introduce the term "ESLs," referring to the stable unmethylated and stable methylated loci. Currently, this definition is only described in the main text, which may cause confusion, particularly since this term "ESL" is introduced in the legend of Fig. 1c without being properly highlighted earlier. It would be helpful to explicitly present this concept in Fig. 1b. Additionally, the number of stable unmethylated and stable methylated loci should be indicated in the figure to provide quantitative context.

We thank the reviewer for this helpful suggestion. We have revised Fig. 1B to explicitly introduce the term ESLs. We have also added the corresponding numbers of stable unmethylated and stable methylated loci to provide quantitative context, as recommended. The figure legend and caption have been updated accordingly for clarity.

- Fig. 1c: The legend states, "Violin plots showing β -values of methylated (top) and unmethylated (bottom) ESLs across 82 cell types." However, it should be clarified that the ESLs plotted here are those identified from the cohort of 1,658 young adults shown in Figure 1a, rather than ESLs independently defined within each cell type or tissue. This clarification is important to avoid the misunderstanding that ESLs were separately called in each dataset.

We agree with the reviewer and have accordingly updated the caption for Fig. 1C:

"Violin plots showing the distribution of β -values for ESLs identified in blood across 82 human cell types using whole-genome bisulfite sequencing data. Top panel: ESLs stably hypermethylated in blood; bottom panel: ESLs stably hypomethylated in blood."

- Line 189: There appears to be a mistake in the statement: "Hypomethylation of normally methylated ESLs was also observed... in Supplementary Figure 2." Based on Supplementary Figure 2, the pattern reflects "hypermethylation" of normally methylated ESLs, not "hypomethylation".

We contend that the term "hypomethylation" is used appropriately in this context. Supplementary Fig. 3 (previously Supplementary Fig. 2) displays ESLs that are typically fully methylated in healthy individuals, with β -values close to 1. In the cancer samples, some of these ESLs show a shift toward lower β -values. Nonetheless, to clarify this point, we have updated the text to replace "hypomethylation" with "reduced methylation" (**Line 209**) and modified the caption of Supplementary Fig. 3 to explicitly define the reference state.

- Line 196: It would be helpful to define "perturbed ESLs" in the main text, not only in the Methods section, since subsequent analyses heavily rely on this concept.

We agree with the reviewer and have added a definition of "perturbed ESLs" to the main text (**Lines 207-208**) to improve clarity for subsequent analyses: "ESLs with β -values above the threshold established from healthy controls (termed perturbed ESLs) ..."

Reviewer #1 (Remarks on code availability):

Appropriate code provided

Reviewer #2 (Remarks to the Author):

We thank Reviewer #2 for their time in reviewing our manuscript as part of the Nature Communications co-review initiative.

Reviewer #3 (Remarks to the Author):

The paper "Blood-Based Epigenetic Instability Linked to Human Aging and Disease" by Basrai S et al studies the biological and clinical significance of a particular age-related DNA methylation pattern, characterized by loci (termed epigenetic stable loci-ESLs) that are stably unmethylated or methylated across over 1000 blood samples from young individuals (all aged 18), but which are demonstrated to undergo destabilization in hematological cancers, and which in other cohorts (e.g. the FHS) is informative of cardiovascular-disease and mortality. An important emerging insight from this study is that the DNA methylation instability (quantified through a measure called DMI) is associated with cardiovascular disease and mortality risk in a manner that seems to be independent of clonal hematopoiesis (CH). High DMI values are also observed in individuals with no clear genetically driven CH, pointing towards other potential sources of CH.

Overall, I think that this is an interesting study, although I do unfortunately have some major serious concerns, some regarding statements about true conceptual novelty, and others of a more technical nature. I now describe these in more detail:

We sincerely thank the reviewer for their careful summary of our study and their overall positive assessment of our manuscript.

Major concerns:

1) Conceptual novelty is questionable: The concept to focus on epigenetically stable loci in a normal reference population, and to then study whether these loci can be informative of disease risk is not entirely novel, as it has been explored in various different contexts. For instance, a 2012 study (Teschendorff et al Genome Med 2012) used differential variance as a means of feature selection to demonstrate that increased DNAm variability can be informative of cervical cancer risk, 3 years in advance of diagnosis. Whilst, the tissue-type and context differs, that approach, termed EVORA, selected for CpGs that are ultra-stable across the normal samples that remained healthy. Other studies have explored the role of stochastic epimutations in blood in the context of large cohorts such as EPIC-Italy, and I can refer the authors to this recent review (Teschendorff & Horvath Nat Rev Genet 2025) where some of these studies were cited. Hence, I would recommend that the authors rewrite the Introduction to more clearly compare the approach taken here to those of previous epigenetic studies, as described above.

We fully agree with the reviewer and appreciate the suggestion to frame our work within a broader context in the field to better highlight its novelty. To address this, we have added the following section to the Introduction, which provides additional crucial background and more directly situates our study relative to existing research (**Lines 78-101**):

"Individuals can show considerable differences between their chronological years and their physiological state, reflecting variability in the rate of functional decline and susceptibility to disease. Among the most widely used biomarkers of this process are epigenetic clocks, which leverage DNA methylation (DNAm) patterns at CpG sites to estimate biological age. These clocks can capture aging-related changes, including associations with morbidity and overall mortality risk²². Clonal hematopoiesis has also been linked to epigenetic age acceleration²³, although correlations are generally weak, consistent with the notion that stochastic CpG epimutations are largely excluded from conventional clocks. Traditionally, epigenetic clocks are built using penalized multivariate regression,

where a function of age is expressed as a weighted linear combination of CpG methylation values. An alternative approach emphasizes DNAm outliers rather than linear associations with chronological age^{24,25}. In this framework, CpGs exhibiting the greatest deviations from standard methylation states are considered informative, offering complementary insight into biological aging processes. This approach has been especially valuable for cancer-risk prediction, with DNAm outliers in preneoplastic tissues marking subclonal expansions with increased malignant potential, as demonstrated in cervical and breast cancers^{26,27}. Beyond oncology and certain neurodegenerative diseases²⁸, the full spectrum of disorders influenced by DNAm outliers and the degree to which these changes are truly stochastic remains unclear.

In this work, we investigate DNA methylation instability (DMI) in blood, examining a broad set of normally stable CpG loci. We find that DNAm variability at those loci increases with age and can inform leukemic burden and cardiovascular risk. We also explore mechanisms by which methylation of these sites may alter cellular function and contribute to disease, offering insights into the biological consequences of epigenetic instability."

2) Technical concerns when identifying ESLs: **(a)** It is a well-known fact that the top PCs of any WB DNAm dataset would correlate very strongly with variations in immune-cell fractions, notably with variations in the neutrophil fraction—the most abundant fraction in WB. Thus, when you search for ESL in WB the most sensible approach would be to focus on loci that are not immune-cell type dependent, say loci that are stably unmethylated <0.2 across all immune cell-types. For these loci, how many neutrophils or lymphocytes there are in any given sample is completely irrelevant, because all underlying cell-types are in the same unmethylated state. Hence, the rationale of the authors to remove samples that display abnormally high or low levels of any given immune cell-type is not well justified. It is completely unnecessary. **(b)** The authors state that "Probes with detection $P > 1 \times 10^{-10}$ in more than 5% of samples were removed using the detection function". This super stringent threshold is insane and completely unjustified. Normally, we would trust DNAm-values that pass a detection P-value threshold < 0.001 . So let us say that a CpG has a detection P-value of $1e-9$ in 10% of the samples and a detection P-value of $1e-11$ in the rest. It is in my opinion not well justified to remove it!!!! It is bad statistical practice and it would be damaging to the community if such errors make their way into published literature. The authors need to redo this. **(c)** It is also very unconventional and a bad idea to divide CpGs up into only 2 states: methylated ($\beta > 0.5$) and unmethylated (< 0.5). In any given cell, DNAm can be in 3 states, and so when we talk about a CpG being unmethylated, one should adopt a much more stringent threshold, for instance $\beta < 0.2$. Likewise, fully methylated would be $\beta > 0.75$ (there is no direct symmetry due to technical assay consideration, hence why the threshold is lowered slightly), with the rest being in a partially or semi-methylated state ($0.2 < \beta < 0.75$). Moreover, later on line 170-172 the authors state that no ESLs displayed intermediate values, but according to the definition given in Methods, there could never be an intermediate state, since the authors used 0.5 as a threshold to call CpGs unmethylated or methylated. It just does not make sense!

a) The reviewer's comment accurately summarizes our approach for identifying stable ESLs with low methylation levels, independent of the proportions of specific blood cell types. Our original intention in removing individual samples from a discovery cohort of young healthy individuals, yet with abnormally count estimates was a precautionary measure, aimed at safeguarding against potential underlying biological anomalies that could inflate methylation variability at certain CpG sites. That being said, we evaluated ESL identification both with and without this filtering step and observed highly similar results. Specifically, the total number of ESLs changed only slightly, from 31,569 to 31,744, corresponding to a percent difference of 0.55%. As insights derived from the downstream analysis remain robust, we agree with the reviewer that this filter is unnecessary. Accordingly, we have removed it from our analysis and updated the manuscript to reflect this change.

b) We thank the reviewer for their comment. To align with standard practice in the field, we re-ran our analysis using a more permissive detection threshold. Consistent with our previous response, this adjustment did not affect the conclusions drawn from downstream analyses.

The manuscript has been updated accordingly (**Lines 125–126**):

"Probes with detection $P > 0.001$ in more than 5% of samples were removed using the detection P function."

c) We wish to clarify that after selecting the 10% least variable CpG sites, none exhibited intermediate mean β -values; in the discovery cohort, all ESLs were either below 0.043 or above 0.962. To better communicate our approach, the previous sentence "Sites with an average β -value < 0.5 were labeled as unmethylated ESLs, while others were categorized as methylated ESLs" has been removed.

The text has been updated to clarify this point (**Lines 135–139**):

"The average β -values at these stable sites showed a pronounced bimodal distribution, with one peak near zero (maximum β -value = 0.043) and another near one (minimum β -value = 0.962). Sites with low β -values were classified as unmethylated ESLs ($n=31,744$, Supplementary Table 2), and those with high β -values as methylated ESLs ($n=6,143$)."

3) ESLs are stable but not ultra-stable: When I inspect Fig.1C, I don't appreciate that these loci are that stable. Whilst the bulk of the beta-values for unmethylated CpGs are indeed < 0.25, I have seen from my own experience, CpGs displaying tighter distributions. Moreover, the violins do extend across the whole range (0 to 1), and it is unclear to me whether this is just an artefact of the plotting routine, or whether indeed there are values in the whole range. Could the authors please clarify this point? For instance, the first blue violin for adipocytes, the density extends from 0 to 1, so does that mean that there are values close to 1 for this otherwise unmethylated ESL? If so, what is the interpretation of these outliers. I note that the same patterns are observed in blood cell-types, which is more relevant.

We would like to emphasize that our aim was to identify stable, but not ultra-stable ESLs. Ultra-stable sites are expected to contribute minimally to the biological associations highlighted in our study, as their signal, by definition, remains largely invariant.

It is also important to note that the ESLs identified in our study were derived from blood samples assayed using methylation arrays. Fig. 1C, by contrast, displays data from whole-genome bisulfite sequencing (WGBS). Differences in data modality are expected to yield distinct methylation distributions, as array-based measurements and WGBS differ in sensitivity and resolution. To further clarify and confirm that the characteristic bimodal methylation pattern of ESLs is observed across diverse human tissues, we refer the reviewer to our response to Reviewer #1, comment #1 and **Figure R1**, where we analyzed healthy samples from kidney, colon, lung, breast, prostate, ovary, and muscle, all assayed with methylation arrays, and compared them to the WGBS samples presented in Fig. 1C. These analyses visually highlight the expected technical differences between the two assay types. Additionally, since Fig. 1C includes data from multiple tissues beyond blood, variation in blood-based ESLs is anticipated. Regarding the reviewer's specific mention of adipocytes and blood cell-types, we observe that the density extends across the full range, including some values near 1 for otherwise unmethylated ESLs. These observations may reflect true methylation at a small number of CpG, but technical factors, such as incomplete bisulfite conversion, read mapping errors, or sequencing artifacts, could also contribute. The key message is that by large, the extreme patterns of hypomethylation or hypermethylation across ESLs are consistent.

4) Biological significance of Fig.3 is unclear: I would like to better understand the biological significance of this figure. If I understood correctly, ESLs that are more perturbed in myeloid malignancies are less accessible in normal myeloid cells, whereas ESLs more perturbed in lymphoid malignancies are less accessible in normal lymphoid cells. So, could the authors go beyond scATAC-Seq, for instance test for chromatin-state enrichment using data from the IHEC/Epigenomic roadmap, as it suggests to me that these ESLs map to repressed regions. If so, what is the biological meaning of increased DNAm at already repressed regions? That most of it is non-functional?

We wish to emphasize that Fig. 3 represents relative chromatin accessibility levels rather than absolute repression, and we have ensured that the term "relative" is consistently used throughout the main text and figure captions. To further underscore this distinction, we have incorporated here the reviewer's excellent suggestion to interrogate data from the IHEC data portal. (Stunnenberg HG et al. Cell. 2016). Comparing the available DNase-seq data from HSCs and T-cells, we confirmed that lymphoid-associated ESL regions exhibit lower accessibility in mature T-cells relative to HSCs (**New Supplementary Fig. 4E,F**). To contextualize these levels, we examined promoters of genes

specifically associated with cells and tissues other than T-cells, and would therefore be repressed. This included KLF1 (erythroid lineage), SPI1 (myeloid lineage), ALB and AFP (liver), MYH7 (muscle), and RBFOX3 and CAMK2A (brain). The DNase-seq signal at these promoters was substantially lower than at the lymphoid-associated ESLs (**Supplementary Fig. 4E**). Similarly, for myeloid-associated ESLs, HSCs exhibited lower relative accessibility than T-cells. Comparison to promoters of genes repressed in HSCs or not relevant for HSCs biology (GATA1, KLF1, PAX5, EBF1, FBP1/2, NOS3) revealed substantially lower accessibility at these inactive promoters than at myeloid ESLs (**Supplementary Fig. 4F**). This observation supports the notion that ESL perturbation occurs at regions that are not fully silenced.

The relevant text (**Lines 230-234**) now reads as follows:

"These contrasting patterns link lineage-associated ESLs, their frequent perturbation in leukemia, and differential chromatin accessibility in corresponding precursor cells, an observation confirmed in a second dataset (Supplementary Fig. 4A-D). We further confirmed these associations using an orthogonal data modality (Supplementary Fig. 4E, F)."

Our results suggest the presence of complementary mechanisms whereby ESL methylation may further reinforce gene repression, or conversely, reduced chromatin accessibility may preserve and stabilize existing methylation marks. Because our findings are correlative, we have kept this interpretation concise in the Discussion. (**Lines 425-428**).

5) Fig.4A seems trivial to me: perhaps I misunderstood, because it is unclear what is being displayed here. But if the x-axis measures for a given ESL, the fraction of samples that are perturbed at this locus, then clearly the average beta-value of that locus over samples will increase since the ESLs are all unmethylated in the normal reference state. Hence, isn't the correlation being depicted in Fig.4A trivial? I don't see what it shows.

We thank the reviewer for raising this important point. We agree that, without further clarification, the message of Fig. 4A may appear trivial. To clarify, the average β -value shown on the y-axis does not represent the mean methylation across all samples, but rather the mean methylation among only those samples in which a given ESL is perturbed (i.e., exceeds the defined methylation threshold). Accordingly, the baseline for this plot is not the unmethylated reference state characteristic of ESLs, but the methylation levels within the subset of samples exhibiting methylation gain.

This analysis reveals that ESLs perturbed in many patients tend to reach higher methylation levels within those affected samples. This pattern may reflect (1) frequent de novo methylation events occurring independently across multiple cells within individuals, and/or (2) the stable propagation and expansion of methylated clones through somatic cell divisions. Our subsequent analysis was designed to investigate the possibility of the latter mechanism. By contrast, if the association were purely trivial, ESLs methylated in only a few samples would display similar or random methylation magnitudes across those samples, resulting in no systematic correlation. To clarify this point, we have revised the Fig. 4A and Supplementary Fig. 5 caption, as well as the text to clarify this interpretation (**Lines 237-240**).

It now reads as follows:

"ESLs that were destabilized in a higher proportion of patients exhibited greater average methylation levels among those affected samples, suggesting that recurrent destabilization is associated with a larger fraction of methylated cells within each patient".

6) SuppFig8: Related to point-4 above, in this figure the authors display around 234 genes whose promoters contain ESLs and which display downregulation with age. However, the original number of genes with ESLs was 1,879 genes, so my question is what happened to the rest of genes (n= 1645)? Did they display overexpression with age or no change? I think this correlative analysis with gene-expression should be expanded: for instance what happens to ESLs that are actually methylated? Do they map to promoters or other gene regions and does their expression change with age? if the claim is that these age-associated DNAm changes at ESLs are consistently associated with changes in gene-expression, that we need a formal global statistical analysis with P-values confirming such a claim.

We wish to clarify that the goal of this analysis was to identify genes with high likelihood of being influenced by ESL perturbation. To this end, we focused on genes containing an ESL within a CpG island near the transcription start site (Fig. 7A,B; new **Supplementary Table 2**).

In response to the reviewer's question regarding "what happens to ESLs that are actually methylated" and the request for a statistical analysis linking ESL methylation to gene expression, we performed an integrated analysis of TCGA data, which includes both DNA methylation and gene expression profiles from the same samples. The large sample size and substantial inter-sample variability across cancer types enabled us to identify genes showing a statistically significant negative correlation between ESL methylation and expression of the associated gene.

To further refine this list, we applied two additional criteria. First, the associated ESL must be frequently perturbed, as infrequent perturbation is less likely to be functionally relevant. Recurrence was determined based on thousands of leukemia samples analyzed as in previous sections of the paper. Second, the gene's expression must be negatively correlated with age. Genes meeting all three criteria, presence of a promoter-proximal ESL, negative correlation between ESL methylation and gene expression, and age-associated downregulation, represent the strongest candidates for functional influence by ESL perturbation and are listed in **Supplementary Table 5**, along with the corresponding statistical measurements.

We have updated **Fig. 7C**, and the corresponding section in the main text (**Lines 338–347**) has been revised to clarify this analytical approach and the rationale for prioritizing genes warranting further investigation.

It now reads as follows:

"To identify genes most likely influenced by ESL perturbation, we focused on three criteria (Fig. 7C): the presence of recurrently methylated promoter-associated ESLs, as infrequent perturbation and lower methylation (Fig. 4A) are less likely to drive meaningful regulatory changes; a negative correlation between ESL methylation levels and gene expression; and a negative correlation between gene expression and age. Age-related genes were identified using data from the Genotype-Tissue Expression project⁵² (Supplementary Fig. 11), while The Cancer Genome Atlas data⁵³, which provides paired methylation and expression profiles across large, heterogeneous cancer cohorts, was used to identify genes for which promoter ESL methylation was associated with lower expression (Supplementary Methods). Genes meeting all three criteria were prioritized for further investigation (Supplementary Table 5)."

7) Significance of motif enrichment is unclear (Fig.7): I also found the results of Fig.7 unclear in relation to what it all means. I understand that increased DNAm at these sites may effect TF-binding, and the authors identify a number of TFs for which binding motifs are enriched among the ESLs. But there is no clear validation of this in relation to any of the identified TFs. Some form of in-silico validation could have attempted. Moreover, I find panel Fig.7C in particular (and to some degree also Fig.7D) to be very unclear. Why do the authors revert to TCGA data? What TCGA cancer-types were included? Did this analysis adjust for cell-type heterogeneity? If anything, panel 7C seems to contradict that authors claim, because I see more datapoints on the positive side of the y-axis indicating a preponderance of positive correlations between promoter DNAm and gene expression?

With respect to the second part of this comment, we direct the reviewer to our previous response to comment #6, where we revised Fig. 7C to reduce potential confusion and modified the text to more clearly convey the purpose of the TCGA analysis.

To investigate the potential effects of methylation at ESLs on transcription factor (TF) binding, we cross-referenced our list of TFs with data derived from methylation-sensitive SELEX, a technique used to determine how cytosine methylation influences DNA binding specificities (Yin et al., Science, 2017). The presence of 5-methylcytosine (5mC) within a TF binding motif can enhance binding, inhibit binding, have context-dependent effects, or exert minimal influence (**Figure R3**). Among the TF motifs located within promoter-associated CpG islands of prioritized age-related genes (Fig. 7E), 113 TFs had been assessed by Yin et al. for 5mC-dependent binding specificity. Of these, approximately

half exhibited reduced binding when CpGs within their motifs were methylated, whereas roughly one-third showed enhanced binding in the presence of 5mC (**Figure R3**). This underscores the complex effects of ESL methylation: methylation can inhibit repressor or activator TFs, enhance binding of activators or repressors, or produce context-dependent outcomes, particularly when multiple methylation events occur simultaneously.

We have added new text (**Lines 370–380**) and a new Supplementary Table (No. 7) describing and referencing this additional analysis.

It now reads as follows:

"To shed more light on the potential role of ESL methylation in modulating TF binding, we cross-referenced promoter-proximal ESLs (Fig. 7E) with TF motifs whose methylation sensitivity has been characterized using methylation-sensitive SELEX⁶¹. This approach allowed us to identify TFs for which the presence of 5-methylcytosine (5mC) within the binding motif can either enhance or inhibit binding in a context-dependent manner (Supplementary Table 7). KLF12 emerged as a compelling example: its binding is enhanced by 5mC, and it functions as a transcriptional repressor^{62,63}. Perturbation of ESLs within KLF12 target promoters (Fig. 7F) could therefore influence gene expression by altering TF occupancy. We emphasize that the impact of ESL methylation on gene regulation is inherently complex. The functional consequence depends not only on whether 5mC enhances or inhibits TF binding, but also on the TF's regulatory role as activator, repressor, or context-dependent modulator, and on combinatorial effects when multiple TFs and ESLs are involved."

Figure R3. Impact of 5-methylcytosine (5mC) on transcription factor binding. Pie chart showing the distribution of experimentally observed effects of CpG methylation on the binding of 113 transcription factors whose motifs overlap promoter ESLs (analyzed in Fig. 7E). Effects are categorized as inhibitory, enhancing, context-dependent, or neutral, highlighting the variable influence of DNA methylation on TF-DNA interactions

8) Adjustment for cell-type heterogeneity when correlating to health outcomes: Related to some of the earlier points, the loci the authors call ESL still display some variation, say on the scale of 0-25% changes in DNAm. I worry that this variation could be driven by unaccounted cell-type heterogeneity, for instance, shifts between the memory and naïve T-cell fractions. Likewise, CHIP has been associated with shifts in immune-cell fractions. Hence, I think it would be extremely important if the authors could demonstrate that DMI correlates with all the health outcomes in a manner that is independent of variations in immune-cell fractions using the higher 12 IC-type DNAm reference panel in the EpiDISH Bioconductor package (see e.g. Luo Q et al Genome Med 2023), which includes memory and naïve T-cell subsets. Indeed, I note that the authors seems unaware of a fairly extensive body of literature demonstrating that naïve T-cell fractions are generally protective of mortality and CVD etc, see e.g. Luo Q et al Genome Med 2023.

With respect to this reviewer comment, we direct the reviewer's attention to Supplementary Fig. 1 and **Figure R1** (related to our response to Reviewer #1, comment 1), which show that variation in DNA methylation instability is not expected to be predominantly driven by any specific cell type, as values remain low across all cell types tested.

We appreciate the reviewer's suggestion and the need for additional, robust evidence. To address this, we applied the EpiDISH package to estimate immune cell-type proportions for all patients analyzed in relation to health outcomes. Hazard ratios were subsequently adjusted for 12 distinct immune cell types to test whether associations between DMI and outcomes were independent of variations in immune cell fractions. Specifically, in the Framingham cohort, the previously observed significant associations between DMI and cardiovascular disease (CVD) and coronary heart disease (CHD) remained significant after adjusting for immune cell fractions. Similarly, in the cardiogenic shock cohort, the effect of DMI on mortality remained significant after accounting for cell-types.

We have updated the relevant Methods section (**Lines 151–153**). The statistical results from these analyses, including hazard ratios and p-values, are now provided in a new **Supplementary Table 4**. In addition, the main text has been revised to reference these supplementary analyses (**Lines 304–309**): "*Previous analyses of health outcomes revealed that higher fractions of naïve CD4⁺ T-cells, naïve B-cells, and NK cells were associated with a reduced risk of all-cause mortality, independent of major epidemiological risk factors and baseline comorbidities³⁸. Cox regression analyses adjusting for the relative abundance of immune cell types⁵¹ confirmed that these associations with CHD and CHF remained significant (Supplementary Table 4).*"

The main text has also been updated at **Line 324** to reflect these supplementary analyses.

Reviewer #4 (Remarks to the Author):

In the manuscript entitled "Blood-Based Epigenetic Instability Linked to Human Aging and Disease" (Manuscript ID: NCOMMS-25-46973-T), Basrai et al. analyzed multiple, diverse cohorts utilizing DNA methylation data generated from the two most recent Illumina platforms: Infinium HumanMethylation450 (450k) and Infinium MethylationEPIC (EPIC) BeadChip arrays. By focusing on probes common to both platforms, the authors define a subset of CpG sites that are typically stable (either consistently methylated or unmethylated) in human peripheral blood, and across multiple tissues, so-called Epigenetically Stable Loci (ESLs). Subsequent analysis demonstrated that these ESLs become epigenetically unstable in hematological malignancies showing aberrant methylation levels. Furthermore, the authors observed that these loci increase variability in methylation with age, and this variability is associated with an elevated risk of developing cardiovascular disease. Based on these observations, the authors developed a new metric called "DNA methylation instability" (DMI), which quantifies the degree of variability at these previously stable CpG sites (especially those that acquire abnormal methylation). They demonstrated that this metric strongly correlates with biological aging and disease progression, suggesting a potential utility for monitoring disease risk or state. Overall, I found the manuscript an interesting paper to read, and the proposed DMI metric holds potential translational interest as it correlates with leukemic burden, aging and cardiovascular risk. While the integration of large-scale methylation datasets, single-cell chromatin accessibility, and multiple cohorts enhances the robustness of the findings, I have raised several issues that should be addressed before this manuscript can be considered for publication. My major concerns are outlined below, followed by several minor ones:

We thank the reviewer for their thoughtful evaluation of our manuscript and for recognizing both its scientific interest and potential translational relevance.

Major points

(1) Given that the study involves multiple datasets generated from two different platforms and distinct cohorts, with a focus on methylation, the authors should provide additional details on the pre-processing of the DNA methylation data. For instance, it is unclear whether the arrays from different cohorts were processed together as a single dataset or separately on a cohort-wise basis. Clarifying the pre-processing workflow, including steps such as quality control, normalization methods or batch effect correction, would strengthen both the rigor and reproducibility of the study. Incorporating exploratory analyses, such as principal component analysis (PCA), is also recommended to assess data concordance across cohorts and platforms. Additionally, since the analysis was limited to CpG

sites shared between the 450k and EPIC arrays, the authors should demonstrate that beta (β) values are consistent across platforms. Differences in probe chemistry or hybridization efficiency between the 450K and EPIC arrays could introduce technical biases that might confound results. A validation step, such as comparing β -values at overlapping CpG sites in a subset of samples profiled on both platforms, would help confirm cross-platform compatibility and therefore, robustness of the reported findings.

We acknowledge the reviewer's point that additional analyses could further confirm cross-platform compatibility. We also wish to emphasize that our data processing approach follows established common practices for analyzing methylation data from multiple generations of Infinium arrays (450K and EPIC), as described by Fortin et al., *Bioinformatics* (2017; PMID: 28035024). Each dataset was processed using single-sample Noob normalization, in which every sample is normalized independently of others, minimizing cross-sample influence and platform-specific bias.

To further substantiate this approach, we followed the reviewer's recommendation and performed principal component analysis (PCA) on separately preprocessed cohorts used in our study to evaluate potential batch effects (**Figure R4**). We analyzed three major groups: (i) individuals without malignancy, (ii) acute myeloid leukemia (AML) cases, and (iii) T-cell acute lymphoblastic leukemia (T-ALL) cases. Each group included data from three or more independent datasets, with both 450K and EPIC arrays represented. For AML and T-ALL, no dataset-dependent biases were apparent. Slight separations were observed among the healthy cohorts, primarily driven by samples from GSE197676, which formed a somewhat distinct cluster. Using this dataset, we observed a correlation between age and DMI. As this observation remained robust across three additional datasets for which age information was available (Fig. 5), we do not consider its inclusion to pose major concerns for the intended analyses.

The corresponding Methods section was revised accordingly (**Lines 119–129**).

"Methylation array data preprocessing

Raw IDAT files were processed using the minfi R/Bioconductor package²⁹. IDATs were loaded via read.metharray.exp into RGChannelSet objects. To ensure cross-platform consistency among different datasets, only probes shared between the 450K and EPIC arrays were retained. Background correction and dye-bias normalization were performed using the preprocessNoob method, which applies single-sample normalization independently to each array³⁰. β -values (0–1 methylation levels) were calculated using getBeta. We then applied additional probe filtering criteria. Probes with detection P > 0.001 in more than 5% of samples were removed using the detectionP function. We also excluded non-CpG (CpH) targeting probes, probes overlapping SNPs (identified with dropMethylationLoci and dropLociWithSnps), probes with previously reported cross-reactivity³¹, age-associated CpGs^{32,33}, and probes on sex chromosomes. A final set of 397,346 probes were considered for further analysis."

Figure R4. Principal component analysis of healthy, AML, and T-ALL cohorts. For each group, PCA was performed on β -values from probes shared between the 450K and EPIC arrays, including individuals from multiple independent datasets assayed on either platform.

To further evaluate potential platform-specific technical differences, we analyzed β -values at ESLs from independent blood samples not included in our primary study. These samples were processed using the same pipeline as all other datasets and assayed on both the 450K and EPIC arrays, revealing no appreciable differences between platforms (**Figure R5**).

Figure R5. Comparison of ESL β -values in blood samples assayed with both the 450K and EPIC arrays. Density plots showing the distribution of ESL β -values. Samples assayed with both platforms were obtained from the GSE86829 and GSE86831 GEO datasets.

(2) The authors report a significant enrichment of ESLs for CpG islands (OR = ?; $p < 0.001$). However, this observation could be influenced by the inherent design of the Infinium MethylationEPIC/450k array, which is biased for CpG-rich regions such as gene promoters and CpG islands (CpGi) as acknowledged in the discussion. As a result, the apparent enrichment for CpGi could reflect the array probe distribution rather than a true biological signal. To strengthen their conclusions, the authors should clarify whether they accounted for this platform-specific bias in their analysis. If not, the observed enrichment may be an artifact of array design. Normalizing their findings against the EPIC array's probe distribution or explicitly addressing further this confounder in the discussion would improve the robustness of their interpretation. The authors may also consider including values corresponding to fold enrichment in the main text.

The reviewer is correct that the Infinium arrays are biased toward CpG-rich regions, such as gene promoters and CpG islands. Nonetheless, our finding of significant enrichment of ESLs in CpG islands remains robust. As shown in Fig. 7A, the left panel displays the distribution of ESLs, while the right panel shows the distribution of all other probes on both the 450K and EPIC arrays included in our study, highlighting differences relative to this background probe distribution.

To clarify this comparison, we have included the odds ratio in the main text (**Lines 334-335**): *"ESLs were significantly enriched within CpG islands relative to non-ESLs (OR = 15.1, $P < 0.001$) ..."*

(3) The authors reported a number of stable CpG sites. However, to provide greater biological insight, it would be important to determine whether these CpGs represent isolated events or whether they cluster within a broader differentially methylated region (DMRs). Regional methylation changes often have stronger functional relevance than single-site alterations. Therefore, the authors should investigate whether these CpGs cluster within DMRs. If DMRs are found, the authors could include illustrative plots of methylation profiles across representative DMRs, which would help to demonstrate the regional patterns of methylation stability observed in the study.

We wish to direct the reviewer to Fig. 7D, which shows a significant increase in promoter methylation in AML compared to CD34⁺ controls across a set of age-related genes whose promoters harbor recurrently perturbed ESLs (defined as methylation exceeding the threshold in >5% of pan-hematological cancer samples). The figure reveals differentially methylated regions in which methylation frequently extends to adjacent non-ESL CpGs within the same CpG islands.

Consistent with this observation, our analysis of whole-genome bisulfite sequencing data from AML patients shows that methylation of recurrently perturbed ESLs frequently coincides with hypermethylation of entire promoter regions. To better illustrate these regional patterns, we have expanded **Supplementary Fig. 12** (originally Supplementary Fig. 9) to include additional representative examples. Additional examples from a single representative patient are highlighted below (**Figure R6**). Lastly, **Supplementary Table 5** was amended to include fold changes (AML compared to CD34⁺ controls) and p-values corresponding to the interrogated DMRs.

Figure R6. Hypermethylation of CpGs adjacent to perturbed ESLs in a representative T-ALL patient. Genomic coordinates and β -values of perturbed ESLs, with surrounding CpGs showing elevated methylation.

(4) The authors validate the consistency of DNA methylation levels at the identified ESLs using whole-genome bisulfite sequencing data from 82 human cell types. To further support the epigenetic stability of these loci, the authors could also examine methylation levels at these ESLs in GTEx samples (PMID: 36510025). This additional analysis would provide complementary evidence of ESL stability across a range of non-hematologic tissues, further reinforcing their findings.

We thank the reviewer for this insightful suggestion to incorporate the GTEx dataset, which significantly strengthens our demonstration of ESL stability across tissue types. The GTEx data are particularly well-suited for this purpose, as they encompass a broad range of normal tissues assayed using the same platform employed in our ESL discovery workflow.

We have now integrated the GTEx results into a **new Supplementary Fig. 2** (panel A) and updated the relevant text (**Lines 189–193**), which now reads as follows:

"Notably, although these ESLs were identified in blood, we were able to detect their characteristic methylation patterns across non-hematological tissues using different profiling technologies (Fig. 1C). While some variation exists, the overall patterns, largely extreme hyper- or hypomethylation across ESLs, are consistently preserved across diverse tissue types (Supplementary Fig. 2A)..."

(5) The authors should provide more comprehensive data. Currently, the supplementary tables only list Probe IDs for the stable CpG sites (and coordinates, in some cases based on the hg19 assembly). This should be supplemented with summary statistics describing the methylation profiles observed across the samples used to establish the thresholds. In addition, the data should include information on the genomic context (e.g., promoter, exon, intron, intergenic) and closest gene to each CpG site as well as, if not present, the genomic coordinates ideally referenced to the hg38 genome build, as this is the most widely used reference genome today. These details should be accompanied by plots illustrating methylation changes, as the current materials do not show the magnitude of changes observed in samples with epigenetic alterations. Including these elements would substantially improve data transparency and interpretability.

We refer the reviewer to **Fig. 2**, which illustrates the magnitude of methylation changes at ESLs across healthy control cohorts compared to a wide range of hematological malignancy cohorts. For transparency and reproducibility, the complete list of ESLs and all datasets used in this study were made publicly available; newly generated datasets have been deposited in appropriate repositories, and all source data are fully accessible.

We appreciate the reviewer's recommendation to provide more comprehensive data. To facilitate data exploration, we have updated Supplementary Table 2 to include detailed annotations for each ESL, as follows:

- Probe ID
- hg38 genomic coordinates
- Summary statistics (mean and SD) of β -values in the discovery cohort
- Recurrence level based on the pan-hematological cancer cohort
- Enrichment in myeloid or lymphoid malignancies
- Closest gene(s)
- Genomic context (e.g., proximity to TSS, exon, UTR).
- Regional context (e.g, CpG islands, shore, etc..)
- Comprehensive Gene-disease associations based on DisGeNET (Piñero J, et al. Nucleic Acids Res. 2017)

(6) The thresholds established to define "perturbed" ESLs were established based on β -values at these CpG sites across three independent cohorts of healthy individuals: GSE124413, GSE132203 and GSE141682. The authors should clearly justify why they used these three particular cohorts for setting such a threshold (99.9th percentile of pooled β -values across ESLs). Currently, these three cohorts are only briefly described in the supplemental Table 1. Potential confounders (e.g., age, sex and ethnicity) could influence baseline methylation levels.

We thank the reviewer for raising this important point regarding potential confounders such as age, sex, and ethnicity. As noted in Supplementary Table 1, we provide references to the original publications for each dataset, which contain additional details on sample characteristics. Notably, these three datasets in particular help to address several key demographic and sample source confounders. Specifically, GSE124413 contains methylation profiles derived from bone marrow, GSE132203 includes peripheral blood samples from the Grady Trauma Project (GTP), a cohort composed primarily of African American participants, and GSE141682 consists of peripheral blood samples from a Chinese cohort. Other variations among the datasets used are described in the Fig. 2 legend.

We have added this clarification to the main text (**Lines 199–203**):

"To enable inter-cohort comparisons, we established a destabilization threshold based on ESL β -values from three independent control cohorts (total $n = 878$), representing baseline methylation levels at these sites (Fig. 2; β -value = 0.05932). These cohorts encompassed diverse ethnic backgrounds, age ranges, and tissue sources, ensuring that the threshold reflects a broadly generalizable background level."

For the healthy datasets analyzed in the present study, the observed differences appear to be minimal (Fig. 2). Therefore, we did not consider it appropriate to focus this particular work on confounder-specific analyses, although future studies using larger datasets should more directly investigate associations between DMI levels and potential confounders such as ethnicity. Instead, we combined several diverse datasets and applied a stringent threshold of β -values across these control samples. We believe that this criterion is robust and broadly applicable across datasets with varying demographic and biological characteristics. Consistent with this approach, we later report DMI associations with health outcomes, in independent datasets, after correcting for key covariates including as age and sex, further supporting the robustness of our findings (Fig. 6, **new Supplementary Fig. 9**).

7) The criteria for defining ESLs appear somewhat arbitrary and lack thorough justification (10% least variable sites across the discovery cohort). A more detailed rationale for this threshold would improve confidence in the robustness of the definition. This concern also applies to the criteria used to identify lineage-specific perturbed ESLs, which similarly appear to be selected without clear justification.

When defining a subset of stable CpG sites from the methylome, we sought to strike a balance between two extremes. Focusing exclusively on ultra-stable CpG sites would, by nature, capture regions that change only slowly over time. Such sites primarily reflect long-term, cumulative alterations, and their slow dynamics may limit the detection of clinically relevant associations. Conversely, prioritizing highly dynamic CpG sites, those exhibiting rapid methylation turnover, could reveal more recent or ongoing methylation events. However, these rapidly changing sites may undergo frequent methylation reversals, diminishing the overall signal. In addition, convergent evolution across independent clonal populations may further reduce their functional interpretability.

We examined the correlation between age and DNA methylation instability (DMI) using two alternative definitions: (i) the 1% least variable unmethylated sites and (ii) all sites with a mean β -value < 0.5 in the discovery cohort. Our results indicated that both focusing exclusively on ultra-stable ESLs and including a broader subset encompassing more dynamic sites (**Figure R7**) were less robust in capturing the DMI–age associations observed with our original site selection (Fig. 5), supporting our original 10% criteria.

Figure R7. Age correlation analysis using different subsets of CpG sites. (A) DMI was calculated as the standard deviation of the 1% least variable CpG sites ($n = 3,401$) in the discovery cohort. (B) DMI was calculated as the standard deviation of all CpG sites with a mean β -value < 0.5 in the discovery cohort ($n = 181,641$). For both analyses, the correlation between age and DMI was assessed across the same four cohorts shown in Main Fig. 5.

Similarly, for the lineage-enriched ESLs, we conducted Fisher's exact tests on the binarized β -values of each ESL (based on the destabilization threshold) to determine whether one group of samples was enriched for destabilization events at a given ESL compared to the other group. Only ESLs with $P < 0.05$ after Bonferroni correction were retained. Here, we repeated the analysis using different cutoff parameters, which yielded inferior results. For example, removing the requirement for a minimum proportion of samples to display an ESL perturbation within any of the "enriched" groups substantially increased the number of lymphoid- or myeloid-associated ESLs. However, the chromatin accessibility profiles across these additional loci were largely similar between the tested fractions with respect to the lymphoid-associated ESLs (panel A), suggesting that this filter effectively excludes low-information sites (**Figure R8**).

While alternative approaches for ESL selection can yield varying strengths of association, optimizing cutoffs may lead to overfitting. Thus, we prioritized consistently validating our observations across multiple datasets to ensure their robustness. Additional data supporting the selection of ESLs, as well as those more frequently perturbed in lymphoid or myeloid cancers, were generated as part of this revision process and are presented in **Figures R1, Supplementary Fig. 2 and Supplementary Fig. 4E,F**.

Figure R8. Lineage-associated ESLs determined using different criteria. (A) To identify lineage-associated sites, Fisher's exact tests were conducted for each ESL without applying a minimum proportion cutoff. All sites with $P < 0.05$ were retained to select lymphoid-enriched ESLs. (B) The same approach was applied to identify myeloid-enriched ESLs.

8) In the survival analysis, participants from the Framingham Heart Study were stratified into "high DMI" and "low DMI" groups based on the cohort-wide median. However, dichotomizing a continuous variable like DMI can reduce within-group variance and may limit the detection of more specific associations with outcomes. As a complement to the dichotomized analysis, the authors could present the full distribution of DMI across participants, either as a supplemental figure or table, to provide a clearer picture of the range and variability in the cohort. Additionally, it would be valuable to repeat the analysis focusing on individuals with extreme DMI values (for example, those below the 10th percentile and above the 90th percentile) to assess whether associations are stronger or more pronounced at the distribution's tails. This approach could help evaluate the robustness and sensitivity of the findings.

We appreciate the reviewer's comment and the opportunity to further evaluate the robustness and sensitivity of our findings. We have added a new supplementary figure showing the distribution of DMI in "low DMI" and "high DMI" individuals (**Supplementary Fig. 9**). Kaplan–Meier and Cox proportional hazards analyses were now performed also using only individuals from the top and bottom quartiles. As expected, these results indicate that associations with clinical outcomes are more pronounced for individuals at the extremes of the DMI distribution.

We have now included these findings in the main text (**Lines 309–312**):

"To assess the impact of extreme epigenetic profiles, we repeated the analyses considering only individuals in the top and bottom quartiles of DMI values. This comparison of extreme quartiles accentuated the differences in clinical outcomes between individuals with high versus low DMI (Supplementary Fig. 9).

9) Following the analysis of ESL methylation versus gene expression and age versus gene expression (and recurrence), 234 genes were selected for further investigation. However, the criteria for determining significance, including whether any multiple testing correction was applied, are not clearly described. The authors should provide these details.

To clarify the criteria for gene selection, we have updated the corresponding section in the main text (**Lines 338–347**).

It now reads as follows:

"To identify genes most likely influenced by ESL perturbation, we focused on three criteria (Fig. 7C): the presence of recurrently methylated promoter-associated ESLs, as infrequent perturbation and lower methylation (Fig. 4A) are less likely to drive meaningful regulatory changes; a negative correlation between ESL methylation levels and gene expression; and a negative correlation between gene expression and age. Age-related genes were identified using data from the Genotype-Tissue Expression project⁵² (Supplementary Fig. 11), while The Cancer Genome Atlas data⁵³, which provides paired methylation and expression profiles across large, heterogeneous cancer cohorts, was used to identify genes for which promoter ESL methylation was associated with lower expression (Supplementary Methods). Genes meeting all three criteria were prioritized for further investigation (Supplementary Table 5)."

In addition, we have updated the relevant Supplementary Methods section, "Identification of ESL-regulated, age-associated genes through integrated methylation, expression, and recurrence analysis," to clarify that Pearson's correlation coefficient was calculated both between methylation and gene expression, and between gene expression and age. We also added a statement regarding significance criteria: "Statistical significance was determined using an alpha of 0.05, with Bonferroni correction applied for multiple hypothesis testing".

10) It is unclear whether the data from the "Differential methylation analysis" included in the supplementary material are provided as a supplementary table. Including such a table could help illustrate the epigenetic changes occurring across these regions. Additionally, Supplementary Figure 9 can be supplemented with further examples (please add methylation scale on y-axis).

In response to the reviewer's comment regarding the WGBS differential methylation analysis (previously Supplementary Fig. 9, now **Supplementary Fig. 12**), we have now included the results of this analysis, fold changes and P-values, as two new additional columns in Supplementary Table 5, along with column descriptions for this table. Additionally, we provide further examples of these genes showing promoter hypermethylation in AML, illustrating that methylation of recurrently perturbed ESLs frequently coincides with hypermethylation of entire promoter regions. Tick marks indicating methylation scale have now been incorporated on the y-axis.

Minor points.

11) Although the study delivers valuable insights into the stability and perturbation of the human methylome across health, disease, and aging, the current title may slightly overstate its scope. Specifically, the title implies a broad investigation of disease-related epigenetic changes, whereas the study primarily focuses on two disease categories: cardiovascular disease and hematopoietic malignancies. To better reflect the manuscript's content, consider narrowing the title to align with the studied conditions.

We appreciate the reviewer's comment on the title. We believe the current title appropriately conveys the study's broader relevance and emphasizes the potential for future studies. Accordingly, we have retained the title as originally submitted.

12) The abstract may benefit from clearly presenting the study's scale through the inclusion of key numbers, such as the exact number of CpG sites found to have consistently stable methylation profiles in healthy blood, the statistical significance thresholds or criteria used for site selection or total number of cohorts analyzed, along with sample size for each cohorts (for example: cardiovascular disease cohort, n = X,XXX; healthy control cohort, n = X,XXX). Presenting these specific numbers would help readers better appreciate the scale and robustness of the work. As currently written, the abstract lacks these critical metrics, making it feel somewhat vague.

We agree with the reviewer and have revised the abstract to include these critical details (**Lines 38–41**):

"Through genome-wide analyses, we identified 31,744 CpG sites exhibiting highly consistent methylation profiles in the blood of young, healthy individuals. We then assessed alterations at these epigenetically stable loci in 8,886 individuals across 29 diverse cohorts, including those with hematological cancers (n = 3,159), cardiovascular complications (n = 2,788), and healthy controls (n = 2,939)."

13) The introduction feels rather brief and concludes abruptly after introducing DMI (Line 80). Adding a paragraph introducing clearly the central research question, the study's hypothesis, or specific objectives, along with a brief description on how the study addresses this question would strengthen this section. Additionally, while the manuscript focus is on epigenetic mechanisms, the introduction focuses predominantly on genetic factors, providing limited discussion of epigenetic dysregulation in aging or clonal hematopoiesis (CH). Expanding the introduction to include a concise overview of previous findings on methylation instability in CH, with references to relevant studies, would offer valuable context and underscore the novelty and importance of the current work. This added background will equip readers with a stronger foundation for understanding the motivation behind the study and allow for a clearer appreciation of the subsequent sections.

We fully agree with the reviewer and appreciate the suggestion to frame our work within a broader context to better highlight its novelty, clarify the motivation, and provide a stronger foundation for readers. To address this, we have expanded the Introduction with corresponding references to discuss the points raised in this comment (**Lines 78-101**):

"Individuals can show considerable differences between their chronological years and their physiological state, reflecting variability in the rate of functional decline and susceptibility to disease. Among the most widely used biomarkers of this process are epigenetic clocks, which leverage DNA methylation (DNAm) patterns at CpG sites to estimate biological age. These clocks can capture aging-related changes, including associations with morbidity and overall mortality risk²². Clonal hematopoiesis has also been linked to epigenetic age acceleration²³, although correlations are generally weak, consistent with the notion that stochastic CpG epimutations are largely excluded from conventional clocks. Traditionally, epigenetic clocks are built using penalized multivariate regression, where a function of age is expressed as a weighted linear combination of CpG methylation values. An alternative approach emphasizes DNAm outliers rather than linear associations with chronological age^{24,25}. In this framework, CpGs exhibiting the greatest deviations from standard methylation states are considered informative, offering complementary insight into biological aging processes. This

approach has been especially valuable for cancer-risk prediction, with DNAm outliers in preneoplastic tissues marking subclonal expansions with increased malignant potential, as demonstrated in cervical and breast cancers^{26,27}. Beyond oncology and certain neurodegenerative diseases²⁸, the full spectrum of disorders influenced by DNAm outliers and the degree to which these changes are truly stochastic remains unclear.

In this work, we investigate DNA methylation instability (DMI) in blood, examining a broad set of normally stable CpG loci. We find that DNAm variability at those loci increases with age and can inform leukemic burden and cardiovascular risk. We also explore mechanisms by which methylation of these sites may alter cellular function and contribute to disease, offering insights into the biological consequences of epigenetic instability."

14) Figure 2B. A more informative representation would be a violin plot combined with an overlaid boxplot. This visualization would provide a more complete view of the data by showing the entire distribution shape plus key summary statistics such as median, quartiles, etc. This combined approach provides much more insight than the current presentation of only the median for each disorder.

We have incorporated a combined violin and boxplot for Fig. 2B, as suggested.

15) Figure 5. I would re-label x-axis to indicate intervals.

Fig. 5 has been updated in accordance with the reviewer's suggestion.

16) Figure 6D. I would recommend removing the table placed to the right of the plot and increasing the plot size for better visualization. The p-values could either be directly incorporated into the plot or color-coded based on significance. The table content could then be moved to the main text or included as a supplementary table.

We removed the table in Fig. 6D as suggested. P values are color-coded based on significance and hazard ratios are indicated in the figure caption.

17) Line 138-139. The authors stated: Clonal hematopoiesis was treated as a binary variable indicating the presence or absence of mutations. Do the authors refer to mutations at specific sets of genes? If so, the authors should mention some genes or cite relevant bibliography.

We have updated the text to clarify (**Lines 159-161**): "Clonal hematopoiesis was treated as a categorical variable, distinguishing individuals harboring somatic mutations in *DNMT3A* or *TET2* from those without mutations"

In addition, we verified that the text in Lines 314–319 provides the necessary supporting context: "We previously performed next-generation sequencing on cardiogenic shock patients and identified significant survival differences between those with clonal hematopoiesis and those without detectable clonal hematopoiesis⁴⁸. To investigate the prognostic value of DMI in these high-risk patients, we profiled a subset of the original cohort, including 29 patients with clonal hematopoiesis driven by common *DNMT3A* or *TET2* mutations, and 35 without clonal hematopoiesis."

REVIEWERS' COMMENTS

Reviewer #1 (Remarks to the Author):

I appreciate the thorough and thoughtful revisions. The point-by-point responses were very clear and the additional analyses, clarifications, and textual modifications appropriately addressed all major concerns raised in the initial review. In particular, the authors have:

- (1) Clarified differences between clonal alterations vs. epigenomic changes in the discussion.
- (2) Added new developmental methylation data and revised interpretations related to early events in malignancy to avoid overstating implications.
- (3) Strengthened statistical methods for clonal preservation analyses.
- (4) Provided helpful annotations linking ESL-associated genes to diseases.
- (5) Adjusted figure legends and terminology for improved clarity and reproducibility, and made numerous careful textual improvements throughout the manuscript.

Overall, the authors have resolved the major conceptual and technical concerns from the initial review. I have no additional concerns and believe this work is appropriate for publication.

We thank the reviewer for their thoughtful feedback and constructive suggestions, which helped improve the manuscript.

Reviewer #2 (Remarks to the Author):

Reviewer #3 (Remarks to the Author):

I thank the authors for thoroughly addressing the concerns I had. I think they did a fantastic job and MS is much improved. There is however just one remaining issue regarding some missing and a wrong citation that should be addressed:

Major points:

1-In the Introduction, when referring to studies that have explored DNAm outliers, there are other blood-based studies that precede Conboy's work, and which have not been cited. For instance, Gentilini D et al Aging 2015 <https://doi.org/10.18632/aging.100792> , Gentilini D et al J Endocrinol Invest DOI: 10.1007/s40618-022-01915-2 .

We agree with the reviewer's suggestion and now reference Gentilini et al., Aging, 2015.

2- The study of Meyer and Schumacher (Ref.25) has absolutely nothing to do with DNAm outliers, so it is incorrect and misleading to cite this work in this context. Please remove it, because stochasticity and outliers are 2 different concepts and it is wrong to confuse the two: outliers could be stochastic (as for instance shown in Teschendorff et al Nat Commun 2016) but they don't have to be, and conversely, many non-outliers can display stochasticity (e.g. there is evidence that the Zhang clock CpGs could be changing stochastically at the single-cell level, Tong H et al Nat Aging 2024) and more generally Tarkhov AE et al Nat Aging 2024).

We have removed the reference to Meyer and Schumacher (2024) from this section to avoid potential conflation of DNAm outliers and stochasticity.

Minor (optional): It should also be noted that historically, biological and clinical relevance of DNAm outliers were first studied in non-blood tissues and that the blood-based studies came later. I am not sure if the relevant sentences in this paragraph should be reordered to emphasize the historical timeline.

We acknowledge this valid suggestion. However, we have chosen to retain the original ordering of the text, as we believe it preserves the conceptual flow of the paragraph and maintains clarity for the reader.

Reviewer #4 (Remarks to the Author):

The authors have provided thorough, point-by-point responses to all of my concerns and have revised the manuscript accordingly. The additional analyses, new figures, and expanded introduction and discussion sections strengthen the manuscript and fully address the issues raised. I therefore recommend this manuscript for acceptance and congratulate the authors on their excellent work.

I have just a couple of very minor comments:

1. In the introduction, both "CpG sites" and "CpG loci" are used. For consistency, I suggest selecting one term and employing it throughout the manuscript.

We now use "CpG loci" throughout the manuscript.

2. The acronym "CpGs" has not been defined. Similarly, the acronym for clonal hematopoiesis (CH) is defined in the Discussion despite this term being used earlier in the text. The authors should consider defining these acronyms when their corresponding terms are first introduced.

We thank the reviewer for this suggestion. "CpG" is a standard, widely recognized acronym in the field, so we have not added a definition. The acronym for clonal hematopoiesis (CH) has now been defined at its first occurrence (line 38).

3. In the methods section, the authors stated that "We also excluded non-CpG (CpH) targeting probes and probes overlapping SNPs." It would be helpful to clarify whether this exclusion applies to all SNPs or specifically to common SNPs (e.g., those with a minor allele frequency > 0.05).

When using minfi's dropLociWithSnps() function, we did not specify a minimum minor allele frequency threshold, which reflects the function's default behavior. We have updated the Methods section to include this detail (line 586).

"We also excluded non-CpG (CpH) targeting probes, probes overlapping SNPs (identified with dropMethylationLoci and dropLociWithSnps, with no minimum minor allele frequency cutoff), ..."